# Neural Networks Decoded: Targeted and Robust Analysis of Neural Network Decisions via Causal Explanations and Reasoning

## Abstract

Despite their success and widespread adoption, the opaque nature of deep neural networks (DNNs) continues to hinder trust, especially in critical applications. Current interpretability solutions often yield inconsistent or oversimplified explanations, or require model changes that compromise performance. In this work, we introduce TRACER, a novel method grounded in causal inference theory designed to estimate the causal dynamics underpinning DNN decisions without altering their architecture or compromising their performance. Our approach systematically intervenes on input features to observe how specific changes propagate through the network, affecting internal activations and final outputs. Based on this analysis, we determine the importance of individual features, and construct a high-level causal map by grouping functionally similar layers into cohesive causal nodes, providing a structured and interpretable view of how different parts of the network influence the decisions. TRACER further enhances explainability by generating counterfactuals that reveal possible model biases and offer contrastive explanations for misclassifications. Through comprehensive evaluations across diverse datasets, we demonstrate TRACER's effectiveness over existing methods and show its potential for creating highly compressed yet accurate models, illustrating its dual versatility in both understanding and optimizing DNNs.

## 1 Introduction

Neural networks have demonstrated transformative potential across various applications, notably image classification (Krizhevsky et al., 2012), medical diagnostics (Esteva et al., 2017), and complex pattern recognition (LeCun et al., 2015), even surpassing humans in certain domains (Silver et al., 2016; Rajpurkar et al., 2017). Yet, their inherent complexity obscures their decision-making processes, turning them into "black boxes" that raise transparency and trust concerns, thus impeding their adoption in sectors requiring explainability, such as healthcare and cybersecurity (Zeiler & Fergus, 2014; Castelvecchi, 2016; Doshi-Velez & Kim, 2017; Lipton, 2018; Papernot & McDaniel, 2018; Zhang et al., 2021). Neural Network Explainability, pivotal in Explainable AI (XAI), aims to clarify DNN decision-making to ensure trust, ethical application, and bias mitigation. Although various XAI strategies have been proposed, including saliency maps (Zhou et al., 2015), Grad-CAM (Selvaraju et al., 2017), LIME (Ribeiro et al., 2016), and SHAP (Lundberg & Lee, 2017), they often present inconsistencies, over-simplification, or architectural constraints, underscoring an ongoing challenge in DNN understanding (Baehrens et al., 2010; Ba & Caruana, 2014; Rudin, 2019).

In this paper, we introduce TRACER, a novel approach based on causal inference theory (Pearl, 2009), to infer the mechanisms through which AI systems process inputs to derive decisions. Recognizing that conventional evaluation metrics based solely on validation datasets may not be indicative of a model's performance in real-world settings and drawing inspiration from Pearl's causal hierarchy, our approach reveals how targeted modifications to input features influence the internal states of neural networks, thereby modelling the underlying causal mechanisms. Specifically, TRACER frames the explainability of neural networks as a causal discovery and counterfactual inference problem, where we observe and analyze all intermediate and final outputs of a model, given any sample, its generated set of interventions, and its counterfactuals. Through the aggregation of multiple such instances, we provide interpretability to state-of-the-art models without requiring any re-training or

architectural changes, thus preserving their performance. In conjunction with an efficient approach for counterfactuals generation, this offers contrastive explanations for misclassified samples, expanding our understanding of not just what happened, but why it happened, and what could have happened under different conditions, thus enabling the identification of potential model blind spots and biases, and addressing the overarching issue of trust. Our main contributions can be summarized as follows:

- We propose TRACER, a framework for estimating the causal mechanisms underpinning DNN decisions, combined with a conditional counterfactual generation method for identifying failure modes, providing actionable insights for improving classifiers.

- We perform comprehensive evaluations of TRACER on image and tabular datasets, providing explanations for correct and misclassified samples, while highlighting its effectiveness in discovering the causal maps that describe the key transformation steps involved in decisions.

- We demonstrate TRACER's versatility in both local and global explainability, as well as its ability to outperform prevalent explanation techniques, identify redundancies in neural network architectures, and aid in the creation of optimized, compressed models.

The paper is structured as follows: Section 2 reviews related work. Section 3 describes the TRACER framework and its foundations. And Sections 4 and 5 present our experimental results and conclusions.

## 2 RELATED WORK

Techniques for DNN interpretability are typically categorized by explainability scope, implementation stage, input/problem types, or output format (Adadi & Berrada, 2018; Angelov et al., 2021; Vilone & Longo, 2021). Early endeavours like saliency maps by Zhou et al. (2015), Grad-CAM (Selvaraju et al., 2017) and Layer-wise Relevance Propagation (LRP) (Bach et al., 2015) visually highlighted key features in input data, but often produced inconsistent or coarse explanations, required structural model changes, compromised performance, or overlooked nuances crucial for true comprehension (Rudin, 2019). Model-agnostic approaches, such as LIME (Ribeiro et al., 2016) and SHAP (Lundberg & Lee, 2017), offer explanations by approximating model decision boundaries but can face challenges like resource intensiveness and inconsistencies in local explanations. Simplifying DNNs for improved interpretability (Che et al., 2016; Frosst & Hinton, 2017) often compromises performance, as simpler models cannot always capture the nuances of complex DNNs. In contrast to the aforementioned methods, rather than merely highlighting influential features, TRACER estimates the causal dynamics that steer DNN decisions, without the need for altering the model or compromising its performance.

Different from associative methods, causal inference techniques probe deeper to uncover cause-effect relationships. The idea of merging causal inference with AI is an emerging perspective, with prior works focusing on causal diagrams and structural equation models to gain such associative understanding (Pearl, 2009; Yang et al., 2019; Xia et al., 2021; Kenny et al., 2021; Chou et al., 2022; Geiger et al., 2022; Kelly et al., 2023). For instance, methods like those proposed by Chattopadhyay et al. (2019), Kommiya Mothilal et al. (2021) and Chockler & Halpern (2024) perform causal reasoning to explain decisions made by image classifiers, focusing on identifying causal elements in the input space, while Reddy et al. (2024) extend this to capture indirect causal effects. TRACER extends causal reasoning deeper into the structure of DNNs, combining causal analysis of both the model's internal workings and the input-output relationships. This enables explainability at both the feature and network-structure level, providing more comprehensive explanations for DNN behavior.

Recent advances have also emphasized counterfactual explanations (Feder et al., 2021), generating hypothetical instances to show how changes in inputs would alter predictions. For example, deep generative approaches using Variational Autoencoders (VAEs) (Pawelczyk et al., 2020; Antorán et al., 2020) or Generative Adversarial Networks (GANs) (Mirza, 2014; Nemirovsky et al., 2022) were proposed to minimize changes to input features to produce counterfactuals. Our approach improves on these by introducing a dual objective that ensures realism through adversarial training while aligning counterfactuals closely with their nearest neighbors in the target class, making them simultaneously plausible and interpretable.

Our proposed approach sets itself apart in two main aspects: (1) rather than only focusing on input features, our approach performs an intervention-based analysis that additionally examines the causal mechanisms within the DNN architecture, identifying how specific layers causally influence the

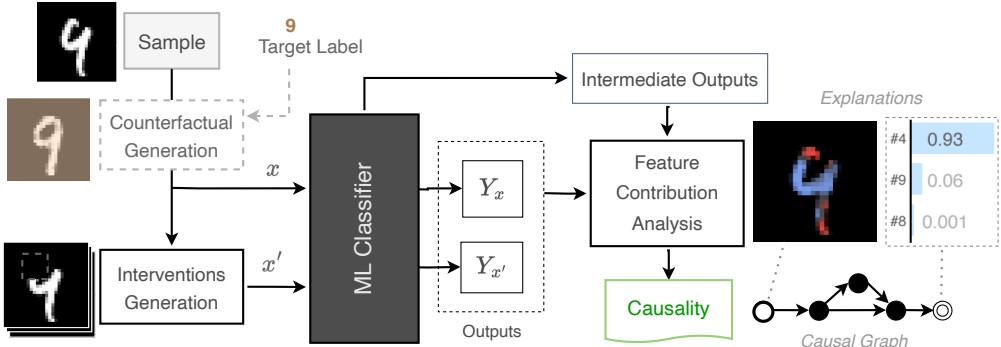

Figure 1: Overview of TRACER with explanations for a misclassification. Interventions and counterfactuals are used to determine the effects of individual features on the models' intermediate and final outputs, leading to the discovery of the mechanisms underpinning the decision-making process.

decision-making process, thereby inferring the critical components (critical layers) within DNNs; and (2) a conditional counterfactual generation method, which synthesizes realistic alternative scenarios to identify model blind spots and biases, while ensuring the generated counterfactuals remain plausible and target specific outcomes through controlled feature changes.

## 3 THEORETICAL FOUNDATIONS AND METHODOLOGY

To understand the internal-workings of DNN architectures, we must consider not only the operations performed by individual layers, but also how they influence one another across the network. TRACER aims to estimate an accurate model of these mechanisms, focusing on the dynamics that govern the network's decisions. Therefore, our methodology, depicted in Figure 1, is structured around:

**Causal discovery.** We analyze the interactions and dependencies within DNNs by systematically altering input features to observe the resulting changes, enabling an effective mapping of the decision pathways. Through this process, we estimate the causal structures that drive the network's decisions, providing a clear understanding of how different features and layers contribute to the outcome.

**Counterfactual generation.** We simulate alternative scenarios by introducing targeted changes to *input features*, allowing us to explore 'what-if' scenarios and observe how specific changes in inputs can lead to different outcomes, providing further insights into the model's sensitivity and robustness.

### 3.1 NOTATIONS

Throughout this paper, we use the following notations to describe our methodology. We denote with $\mathbf{X}$ the input features and $\mathbf{Y}$ the output or prediction made by a DNN. A single input instance is denoted as $\mathbf{x} \in \mathbb{R}^d$, where $d$ is the input dimensionality. The function $f$ represents a neural network, mapping inputs to outputs. We denote the output of a specific layer or group of layers in the network as $\mathbf{g}_i$. For causal discovery, $\mathrm{do}(X = x_j)$ represents an intervention where the value of $X$ is set to $x_j$, and $P(\mathbf{Y} \mid \mathrm{do}(X = x_j))$ describes the probability of $\mathbf{Y}$ given this intervention. To measure the similarity between the outputs of two layers, we use the Centered Kernel Alignment $\mathrm{CKA}(K_i, K_j)$, where $K_i$ and $K_j$ are the kernel matrices corresponding to layers $i$ and $j$. A binary matrix $\mathbf{B}(K_i, K_j)$ indicates similarity between layers, with elements $b_{i,j} = 1$ when $\mathrm{CKA}(K_i, K_j)$ exceeds a threshold.

### 3.2 PRELIMINARIES

Causal theory provides the means to model cause-effect relationships, offering a departure from mere observational statistics to tackle questions about interventions and counterfactuals (Pearl, 2009). To this end, the language of Structural Causal Models has been proposed to formalize these relationships.

**Definition 1** (Structural Causal Model). A Structural Causal Model (SCM) $\mathcal{M}$ is a 4-tuple $(U, V, \mathcal{F}, P(U))$, where $U$ is a set of exogenous variables determined by factors external to the

model; $V = \{V_1, V_2, \ldots, V_n\}$ is a set of endogenous variables, each influenced by variables within the model; $\mathcal{F} = \{f_1, f_2, \ldots, f_n\}$ is a set of functions, each $f_i$ mapping a subset of $U \cup V$ to $V_i$; and $P(U)$ is a probability distribution over $U$. For every endogenous variable $V_i$, its value is determined by $V_i = f_i(\text{pa}(V_i), U_i)$, where $\text{pa}(V_i)$ represents the parents or direct causes of $V_i$, and $U_i \subseteq U$.

Pearl's Causal Hierarchy (PCH), grounded in SCMs, further refines our understanding by categorizing causal knowledge into three distinct levels, which serve as TRACER's foundations[1]:

**1. Association.** We extract dependency structures from the DNN activations and outputs $P(Y^{(i)} \mid X)$, where $X$ and $Y^{(i)}$ represent the input and the $i$-th layer's output variables, respectively;

**2. Intervention.** By selectively manipulating feature values, we estimate the intervention distributions $P(Y^{(i)} \mid do(X = x_j))$ to understand the effect of particular features on the final decision[2];

**3. Counterfactual.** We explore alternative (or hypothetical) input scenarios and compute the counterfactual distributions, $P(Y^{(i)}_{X=x'} \mid X = x)$, which quantifies the model's output distribution if a certain input were set to a particular value, given that we actually observed another input.

By identifying how specific input features and intermediate layer activations influence the model's final predictions, TRACER provides a unique approach for capturing an abstract overview of the distinct computational components driving DNN decisions. This structured approach allows us to produce explanations that clarify both the direct influence of features and how the model's predictions would change under different input conditions.

**Definition 2** (Explanation). Given a $d$-dimensional input $\mathbf{X} = \mathbf{x} \in \mathbb{R}^d$, an explanation for the output $y$ of a model $\mathcal{F}$ is a masked input $\mathbf{x}_E = \mathbf{x} \odot \mathbf{M} \in \{0, 1\}^d$ for which the following conditions hold:

- $\Gamma 1$ **(Correctness):** The model $\mathcal{F}$, when evaluated on the input $\mathbf{x}$, produces the output $y$.
  $(\mathcal{F}, \mathbf{x}) \models (\mathbf{X} = \mathbf{x})$ and $\mathcal{F}(\mathbf{x}) = y$, where $\models$ denotes logical entailment.

- $\Gamma 2$ **(Sufficiency):** There exists a *mask* $\mathbf{M} \in \{0, 1\}^d$ such that the resulting explanation $\mathbf{x}_E = \mathbf{M} \odot \mathbf{x}$ produces the same output as the original input: $(\mathcal{F}, \mathbf{x}_E) \models \mathcal{F}(\mathbf{M} \odot \mathbf{x}) = \mathcal{F}(\mathbf{x}) = y$. This condition ensures that the features selected by $\mathbf{M}$ are sufficient to explain $y$. Let a mask $\mathbf{M}'$ be defined such that the set of active features in $\mathbf{M}'$ (i.e., where $\mathbf{M}'_i = 1$) is not a subset of those in $\mathbf{M}$. Formally, $\{i : \mathbf{M}'_i = 1\} \not\subseteq \{i : \mathbf{M}_i = 1\}$, then $(\mathcal{F}, \mathbf{x}'_E) \models \mathcal{F}(\mathbf{M}' \odot \mathbf{x}) \neq y$.

- $\Gamma 3$ **(Minimality):** The mask $\mathbf{M}$ is minimal, meaning that no strict subset of active features in $\mathbf{M}$ suffices to produce the same output. Formally, for every mask $\mathbf{M}' \in \{0, 1\}^d$ such that $\{i : \mathbf{M}'_i = 1\} \subset \{i : \mathbf{M}_i = 1\}$, the masked input $\mathbf{x}'_E = \mathbf{M}' \odot \mathbf{x}$ is insufficient to produce the same output: $(\mathcal{F}, \mathbf{x}'_E) \not\models \mathcal{F}(\mathbf{M} \odot \mathbf{x}) = y$.

Note that *(i)* in this definition, $y$ can be set to any specific label to produce explanations for misclassifications or rare events; and (ii) partial explanations can be simplified to binary decisions (i.e., whether a feature is relevant or not) when computing feature attributions (defined in Section 3.4).

## 3.3 CAUSAL DISCOVERY

To discover a faithful representation of the causal mechanisms underpinning DNN models, we perform an intervention-based analysis where we systematically change the values of input features and study the effects on a given classifier. By observing the internal states and outputs of the classifier, we can deduce how specific components contribute to the final decisions, offering an understanding of the model's causal structure and enabling the identification of key layers or connections that highly influence the model's predictions. Furthermore, by collecting the observed effects of all interventions, we establish an abstract causal map to visualize the interplay between different network components, and asses their collective influence on the DNN outputs. Ultimately, the insights gathered from our approach enable the debugging and refinement of neural network classifiers.

### 3.3.1 INTERVENTIONS

In our analysis, interventions are crucial for isolating and understanding the causal significance of specific input features. Given an input vector $x \in \mathbb{R}^d$, where $d$ denotes the dimensionality of the input

---

[1]Our analysis levels follow the terminology from Pearl's Ladder of Causation (Pearl & Mackenzie, 2018)

[2]$do(\cdot)$ denotes the do-operator, as defined by Pearl (Pearl, 2009).

space, an intervention is simulated by replacing a subset of $x$ with a predetermined baseline value $b$. For a specified subset of indices $I \subseteq \{1, \ldots, d\}$ corresponding to the features under intervention, the intervened features are given by: $x_i' = b \cdot \mathbb{1}\{i \in I\} + x_i \cdot (1 - \mathbb{1}\{i \in I\})$, where $\mathbb{1}\{i \in I\}$ indicates 1 when $i$ is in the set $I$ and 0 otherwise. Assuming $b$ to be causally independent (e.g., binary mask), all input features, before and after interventions, can be considered exogenous variables in the causal map due to their values being set externally and not being influenced by other variables in the model.

**Proposition 1** (Causal Isolation of Intervened Samples). *Let $F : \mathcal{X} \rightarrow \mathcal{Y}$ denote the mapping function of a DNN. For any $x \in \mathcal{X}$, $I \subseteq \{1, \ldots, d\}$, and $b \in \mathbb{R}$, the intervened sample $x'$ isolates the causal effect of the features in $I$ on $F$ by setting the values of $x_i, \forall i \in I$ to $b$. (Proof in Appendix A.1)*

By performing such interventions, we effectively isolate and examine the causal impact of specific features on the output, allowing us to determine which features are causally pivotal for the model's decisions, and to measure the depth of their influence. The value chosen as baseline can carry significant importance in our intervention framework. In cooperative game theory, Shapley values (Shapley et al., 1953) use baselines to evaluate each player's contribution by averaging their marginal impacts across all possible coalitions. This notion has been adapted for interpreting machine learning models (Lundberg & Lee, 2017), inspiring our use of baselines as neutral points of reference. In our approach, the baseline aims to counteract or neutralize the impacts of altered features, isolating the original input's influence on the output without the bias introduced by those features. By contrasting the results from such intervened inputs with the original's, we extrapolate the causal relationships between input features and model outputs.

### 3.3.2 CAUSAL ABSTRACTION

Given an input sample and its interventions, TRACER collects the intermediate and final outputs of the classifier to perform a focused comparison of representations across network layers and extrapolate an accurate estimation of the causal dynamics driving the network's decisions. For this analysis, we use Centered Kernel Alignments (CKA), a prevalent approach for quantifying similarities between high-dimensional embeddings (Kornblith et al., 2019). Let $f_i \in \mathbb{R}^{n \times d_i}$ and $f_j \in \mathbb{R}^{n \times d_j}$ denote the activations of two distinct layers in a network for a set of $n$ input samples, where $d_i$ and $d_j$ represent the dimensionalities of the activations for layers $i$ and $j$, respectively. Their respective kernel matrices are defined as $K_i = f_i f_i^T \in \mathbb{R}^{n \times n}$ and $K_j = f_j f_j^T \in \mathbb{R}^{n \times n}$ to obtain their CKA similarity:

$$\text{CKA}(K_i, K_j) = \text{HSIC}(K_i, K_j)/\sqrt{\text{HSIC}(K_i, K_i) \times \text{HSIC}(K_j, K_j)},$$

where $\text{HSIC}(K_i, K_j)$ is the Hilbert-Schmidt Independence Criterion (HSIC) for the kernel matrices, and given by $\text{HSIC}(K_i, K_j) = (n-1)^{-2} \text{Tr}(H K_i H K_j)$. Here, $H$ is a centering matrix given by $H = I - \frac{1}{n}\mathbf{1}\mathbf{1}^T$, with $n$ being the number of samples, $I$ the identity matrix, and $\mathbf{1}$ a vector of ones. $\text{Tr}(\cdot)$ denotes the trace of a matrix.

The use of CKA for evaluating representation similarity offers several advantages, including: *(i)* **Normalization:** CKA scores range from 0 (completely dissimilar) to 1 (identical), allowing straightforward comparison across layers; *(ii)* **Flexibility:** It accommodates various kernel functions, such as linear or Gaussian, enabling flexibility based on specific requirements of the analysis; and *(iii)* **Robustness:** The use of kernels allows CKA to operate in a richer feature space, providing a more comprehensive similarity measure.

Upon obtaining the similarity measures, we establish causality by grouping layers based on their CKA values, where we create a binary matrix $\mathcal{B}(K_i, K_j)$, which is defined as $\mathcal{B}(K_i, K_j) = 1$ if $\text{CKA}(K_i, K_j) \geq 1 - \epsilon$, and 0 otherwise, with $\epsilon$ representing a predetermined threshold that defines the maximum acceptable dissimilarity for two layers to be considered functionally similar and grouped into a single causal node. While a smaller $\epsilon$ (i.e., stricter similarity criteria) leads to more granular grouping, a larger $\epsilon$ results in broader grouping. For our causal analysis, such similarity suggests that these layers contribute to a shared causal node representing an endogenous variable and describing a distinct structural equation in our causal model.

**Definition 3** (Layer Groups). Let $F(x) = f_k \circ \ldots \circ f_1(x)$ denote the compositional form of the neural network classifier, with $f_i$ representing the $i$-th layer of the network. And let $\mathcal{B}$ denote the binary CKA matrix. Two distinct layers $f_i$ and $f_j$ are said to belong to the same layer group $g_l$ if and only if $|i - j| = 1$, $\mathcal{B}(K_i, K_j) = 1$, and $\forall k > l$, $f_i, f_j \notin g_k$. All layer groups are mutually exclusive and collectively exhaustive, i.e., $\forall p \neq q$, $g_p \cap g_q = \emptyset$, and $\bigcup_{i=1}^m g_i = \{f_1, f_2, \ldots, f_k\}$.

**Theorem 1** (Layer Grouping). *Let a sequence of layers $\{f_j, f_{j-1}, \ldots, f_i\}$ within a neural network $F(x)$ be classified under the same Layer Group, i.e., $\mathcal{B}(K_j, K_{j-1}) = \mathcal{B}(K_{j-1}, K_{j-2}) = \ldots = \mathcal{B}(K_{i+1}, K_i) = 1$, where $\mathcal{B}(K_i, K_j) = 1$ if $CKA(K_i, K_j) \geq 1 - \epsilon$. The collective causal influence of this sequence on $F$'s output is encapsulated by a single composite layer $g_{ij}$: $F'(x) = f_k \circ \ldots \circ g_{ij} \circ \ldots \circ f_1(x)$, where $g_{ij} \equiv f_j \circ f_{j-1} \circ \ldots \circ f_i \equiv f_i$. (Proof in Appendix A.2)*

This definition of "Layer Groups" aggregates layers into cohesive groups, where each group estimates a distinct node in the decision mechanism of the network. Through this aggregation, we effectively abstract the composition of layers into single causal nodes when their computations are found to be redundant, allowing for a more streamlined and high-level understanding of the network's processes.

**Theorem 2** (Necessary and Sufficient Conditions for Causal Nodes). *Let $F : \mathbb{R}^n \to \mathbb{R}^m$ be a DNN defined by composition as $F = f_k \circ \ldots \circ f_1$ where each $f_i : \mathbb{R}^{d_{i-1}} \to \mathbb{R}^{d_i}$ represents the transformation applied by the $i$-th layer, $d_0 = n$, and $d_k = m$. Let $g = g(r) = \ldots = g(s) = \{f_i\}_{i=r}^s$ with $1 \leq r < s \leq k$ be a subset of consecutive layers. $g$ constitutes a causal node as per Definition 3 if and only if $\forall i \in \{r, \ldots, s-1\}, CKA(K_i, K_{i+1}) \geq 1 - \varepsilon$, where $K_i$ is the kernel matrix of layer $i$ and $\varepsilon \in (0, 1)$ is a predefined similarity threshold. (Proof in Appendix A.3)*

**Definition 4** (Causal Links between Layer Groups). Let $g_a$ and $g_b$ denote two distinct layer groups within a neural network. A causal link between $g_a$ and $g_b$ is established if they are adjacent, i.e., $\exists f_i \in g_a, \exists f_j \in g_b$ such that $|i - j| = 1$, or there exists a causal connection between layers in $g_a$ and $g_b$ either directly, where $\mathcal{B}(K_i, K_j) = 1$ for some $f_i \in g_a$ and $f_j \in g_b$, or indirectly through an intermediate group $g_{a+1} = g_c = g_{b-1}$ via a layer $f_k \in g_c$ satisfying $\mathcal{B}(K_i, K_k) = 1$.

By adopting definitions 3 and 4, which ensure that layer groups are mutually exclusive and non-overlapping, we capture the internal dependencies of DDNs, leading to the discovery of layer-wise abstractions that describe the structural equations governing our causal model. This enriched perspective allows for more powerful explanatory modelling through better understanding of the interplay between layers, and how they collectively shape the network's decisions. Consequently, our approach offers valuable insights into the high-level causal mechanisms that shape the network's behavior, and allows us to provide an abstract, structured, and interpretable view of the causal dynamics that are intrinsic to its operations.

## 3.4 ESTIMATION OF CAUSAL EFFECTS

We define the *Average Causal Effect* (ACE) to quantify the causal impact of interventions on the network's outputs, quantitatively capturing both their direction and magnitude.

**Definition 5** (Average Causal Effect). Let $g_i'(x) = \text{softmax}(g_i(x))$ and $g_i'(x') = \text{softmax}(g_i(x'))$ denote the normalized outputs of a Layer Group $g_i$ for a given input $x$ and its intervention $x'$. The normalization of these outputs is performed to transform the activation scores into valid probability distributions, with which the Average Causal Effect (ACE) can be defined as the expected value of the product of the signed Kullback-Leibler (KL) divergence between their probability distributions:

$$\text{ACE}_i = \mathbb{E}_{P(X)}\left[|\Delta_x^i| \cdot \text{KL}\big(P(g_i'(x) \mid do(X = x')) \| P(g_i'(x) \mid do(X = x)))\big)\right],$$

where $\Delta_x^i = g_i'(x) - g_i'(x')$ represents the sign of the change induced by the intervention, and $\text{KL}(\cdot)$ represents the KL divergence quantifying the changes between the probability distributions.

This definition provides a robust estimation of the causal effects, allowing us to understand how (direction: positive versus negative contribution) and by how much (magnitude) specific interventions influence the outputs.

**Remark.** *Any intervention that produces outputs sufficiently similar to those produced by the original input has little to no impact on the Average Causal Effect.*
*If the intervention on input $x$ to produce $x'$ results in minimal change in the output of a Layer Group $g_i$, such that $g_i'(x) \approx g_i'(x')$, then with all other features of $x$ remaining untouched, the change induced by $x'$ approaches 0, leading to minimal or negligible contribution. Formally, if $g_i'(x) \approx g_i'(x')$, then:*

$$KL\big(P(g_i'(x) \mid do(X = x')) \| P(g_i'(x) \mid do(X = x)))\big) \approx 0 \implies CE_i = 0.$$

This suggests that interventions which do not substantially alter the output of a Layer Group have a negligible causal impact on the model's output, as measured by the ACE. Our approach henceforth consists of generating interventions, such that those with no effect according to our definition above, are considered not part of the explanation.

### 3.5 COUNTERFACTUAL GENERATION

To improve classification performance and mitigate biases, we explain misclassified samples through a TRACER analysis of counterfactuals, identifying specific feature changes that should be applied to samples to obtain the desired outputs. Counterfactuals, defined as hypothetical data instances that, if observed, would alter the model's decision, must be valid and plausible. To generate such instances, we use generative models like Generative Adversarial Networks (GANs) (Goodfellow et al., 2020) and include these constraints into our training process. Specifically, we propose a novel plausibility constraint, whereby the counterfactual generators are trained using both adversarial training to ensure realism, and a proximity-based regularization term to enforce similarity between the generated counterfactuals and real instances from a target class. This results in realistic counterfactuals requiring minimal changes to the original data. While our proposed constraints can be adapted to various types of generative models (e.g., VAE, GAN, normalizing flows), the model we discuss hereinafter assumes an autoencoder-based GAN architecture.

Given an input $x \in \mathbb{R}^d$ and a target output $y^*$, our GAN-based counterfactual generation model is defined such that the generator uses an encoder function $E_x$ to map the input $x$ to a condensed latent representation $z_x = E_x(x)$. The desired model output $y^*$, typically an integer label, is transformed into a one-hot encoded vector $o(y^*) \in \mathbb{R}^k$, where $k$ is the number of classes, using the Kronecker delta function $o_i(y^*) = \delta_{iy^*}$ for $i = 1, \ldots, k$. This latent representation $z_x$, concatenated with the one-hot encoded target label $o(y^*)$ to form an augmented latent vector $z = [z_x; o(y^*)]$, is processed by the decoder $D$ to generate a counterfactual instance $x^* = D(z)$. To verify the authenticity of the generated counterfactual $x^*$, the discriminator $\mathcal{D}$ evaluates whether $x^*$ appears realistic and plausible by distinguishing between original data samples and those produced by the generator.

The GAN is optimized using a dual objective: *(1)* ensure the authenticity of the generated counterfactual $x^*$ and *(2)* maximize its similarity with its nearest neighbour $x_{\text{nn}}$ among real samples of its training dataset whose label correspond to some target class. Specifically, we combine the conventional GAN loss and a proximity measure $d(x^*, x_{\text{nn}})$, with $\lambda$ as the balancing coefficient: $\mathcal{L} = (1 - \lambda)\,\mathcal{L}_{\text{GAN}} + \lambda\,d(x^*, x_{\text{nn}})$, ensuring that generated counterfactuals remain minimally different from real instances in the target class, thereby preserving plausibility while leading to the desired prediction. This approach offers a flexible and data-efficient process that closely aligns the generated counterfactuals with the actual data distribution, while conditioning on priors for controlled outputs.

**Remark.** *The regularization distance $d(x^*, x_{nn})$, essential for maintaining plausibility, can be implemented using metrics such as $\ell_1$, $\ell_2$, or perceptual loss. By introducing perturbations $\delta_i$ to the latent representation $z_x$ before decoding, training with this regularization enables the generation of multiple distinct plausible counterfactuals $x^* = D([z_x + \delta_i; o(y^*)])$, thereby reducing mode collapse. We choose in our experiments the $\ell_1$ norm as regularization metric to encourage sparsity in the differences between $x^*$ and $x_{nn}$, promoting minimal and interpretable changes to the original input.*

Employing generative models for counterfactual generation, rather than relying on nearest neighbors during inference, offers several advantages. First and foremost, relying on real data points as counterfactuals would require storing large datasets, potentially leading to memory constraints. This could be particularly problematic in applications where storage is expensive or limited. To address this, we train the counterfactual generator on a small random subset of the training set (e.g., 10%), which is afterwards discarded, eliminating the need for storage. Moreover, this allows us to generate plausible, novel counterfactuals on-the-fly, avoiding computational costs and latency associated with dataset searches, while enabling broader exploration of the feature space.

## 4 EXPERIMENTS AND RESULTS

In this section, we evaluate our proposed explainability method, TRACER[3], primarily emphasizing its causal discovery facets. We perform our initial experiments using the well-known MNIST (Deng, 2012) and ImageNet (Deng et al., 2009) datasets, which are standards in image classification tasks, and on the CIC-IDS 2017 (Sharafaldin et al., 2018) network traffic dataset to demonstrate TRACER's applicability to tabular datasets. We use a modified AlexNet (Krizhevsky et al., 2012) and the ResNet-50 (He et al., 2016) architecture as our MNIST and ImageNet classifiers, respectively. Since AlexNet

---

[3]The source code of TRACER will be made available on GitHub upon publication of the paper.

is originally designed for ImageNet classification, we modified it to take as input single-channel images of size 28×28 pixels, and changed the output layer to have 10 units corresponding to the MNIST digit classes. The base models were then trained on the entire training sets before analysis with TRACER. The training parameters are described in Appendix F.

## 4.1 CAUSAL DISCOVERY AND FEATURE ATTRIBUTIONS

To evaluate TRACER's effectiveness in uncovering the causal pathways that govern DNN decision-making processes, the relationships between activations of different layers are analyzed using CKA similarities. This involves comparing activations from the original input with those from its interventions, to identify their influence on the decisions. We perform interventions by systematically masking pixels in a grid pattern, setting pixel values to zero before normalization. For MNIST, the interventions applied consist of 3×3 patches with a 1×1 sliding window. As depicted in Figure 2, TRACER discerns layer groups forming causal nodes and identifies the causal links between them. Specifically, eight activation outputs from the MNIST classifier are observed and analyzed, revealing inherent groupings based on similarity patterns across the network layers. This observation has led to the identification of four distinct causal nodes, with the lack of causal connections between non-adjacent layer groups indicating a *linear causal chain* driving the model's decision for the given sample.

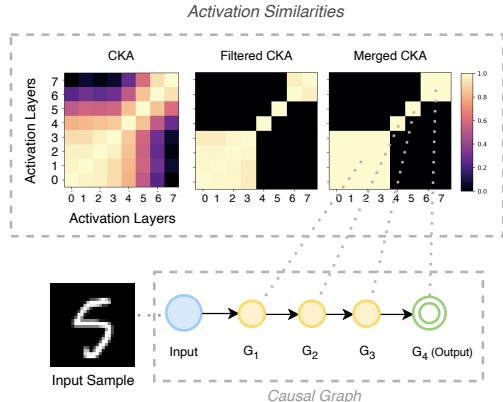

Figure 2: TRACER's causal analysis results for an MNIST sample classified by AlexNet. The causal structure is inferred using CKA similarities between activation outputs from various layers. Nodes in the resulting causal graph symbolize layer groups, while the connections between them capture their causal relationships.

To quantitatively assess the reliability of TRACER, we measure how often a given model's predictions remain consistent when key features identified by our approach are randomly perturbed.

Formally, let $f$ be the classification model. For a dataset $X$, each sample $x \in X$ is coupled with an explanation mask $M(x) \in \{0,1\}^d$ generated by an explainability method, where $M(x)_i = 1$ indicates that the $i$-th feature of $x$ is significant. Let $\mathcal{P}$ denote a perturbation function which modifies $x$ by targeting a proportion $p$ of the significant regions of the explanation. This perturbation function $\mathcal{P} : \{0,1\}^d \times [0,1] \to \{0,1\}^d$ can be defined as follows to produce a binary perturbation matrix:

$$\mathcal{P}(M(x), p) = 1 - \mathbb{1}\{i \in S(M(x), p)\}, \text{ where the set } S(M(x), p) \subseteq \{i \in \{1, \ldots, d\} : M(x)_i = 1\} \text{ is defined as } S(M(x), p) = \{i_1, i_2, \ldots, i_k\} \text{ with } k = \lfloor p \cdot |M(x)| \rfloor, |M(x)| = \sum_{j=1}^{d} M(x)_j, \text{ and } i_1, i_2, \ldots, i_k \text{ are sampled uniformly at random from the indices } \{i \in \{1, \ldots, d\} : M(x)_i = 1\}.$$

Thus, $\mathcal{P}(M(x), p)$ produces a perturbation matrix for exactly $k$ significant features of $x$, leaving all other features unchanged. With $x' = x \odot \mathcal{P}(M(x), p)$ describing the perturbed sample, the reliability score for the explanations can be obtained as: $S = |X|^{-1} \sum_{x \in X} \mathbb{1}\{f(x) \neq f(x')\}$, where $|X|$ is the number of samples in the dataset, and $\mathbb{1}\{\cdot\}$ is an indicator function returning 1 if the predictions before and after applying the explainability mask differ.

This score captures the sensitivity of the model's predictions to changes in areas deemed critical by the explainability method, thereby providing insights into the reliability of the explanations generated. To assess the robustness of TRACER and compare its per-

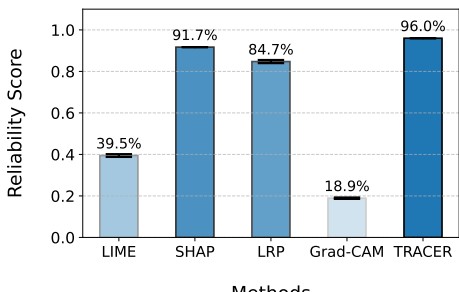

Figure 3: Reliability scores of different explainability methods on the MNIST dataset.

formance against that of existing explainability methods, we use this reliability metric on explanations produced by the different approaches when evaluated on all test samples of the MNIST dataset. The results, depicted in Figure 3, show the average and standard deviation of each method's scores

over 10 trials, for a perturbation factor $p$ of 0.5, demonstrating TRACER's superior performance and consistency in producing meaningful and reliable explanations.

## 4.2    GENERALIZATION AND SCALABILITY

In this experiment, we highlight the broad adaptability of our approach across various neural network architectures and datasets. To this end, we evaluate TRACER on the ImageNet dataset, as well as on a Network Intrusion Detection dataset, explaining the decisions of both simple and complex NN architectures such as MLP and ResNet-50. Intervention applied on ImageNet samples occlude parts of the inputs (setting them to zero before normalization), with $7 \times 7$ patches and a $3 \times 3$ sliding window. And for the Network Intrusion Detection dataset, we use $1 \times 3$ patches with a $1 \times 1$ sliding window.

Given the wide variety and realisic nature of the samples in the ImageNet dataset, its classification results with the ResNet-50 architecture provide a solid benchmark for highlighting the limitations of existing explainability methods and comparing their performances to that of TRACER. For this comparison, we selected LIME, SHAP, LRP, and Grad-CAM as benchmarks, since they are among the most widely adopted and representative explainability methods in the literature. The results, depicted in Figure 4 show that while existing methods struggle to produce consistent explanations, TRACER provides coherent and comprehensive explanations that highlight the most important features and patterns that drive the classifier's decisions. Further comparison of these methods, discussed in Appendix D.1, highlight more distinctions between TRACER and existing methods, particularly when using DNN architectures that exhibit complex interactions.

Diving deeper into the versatility spectrum, we challenge TRACER with the intricacies of structured data using the CIC-IDS 2017 network traffic dataset. This dataset, reflecting authentic network dynamics, unfolds a distinct set of challenges useful for evaluating explainability methods (e.g., diverse data types and intertwined correlations). For example, in an instance where a DDoS-attack-induced traffic is erroneously classified as benign (see Appendix D.2), TRACER identifies and elucidates features emblematic of the attack through its causal analysis. Specifically, TRACER reveals that features such as port numbers and data transfer dynamics are essential for the detection of such threats. Overall, the granularity and transparency of explanations provided by TRACER, especially in domains such as cybersecurity, accentuate its potential to build trust in critical applications.

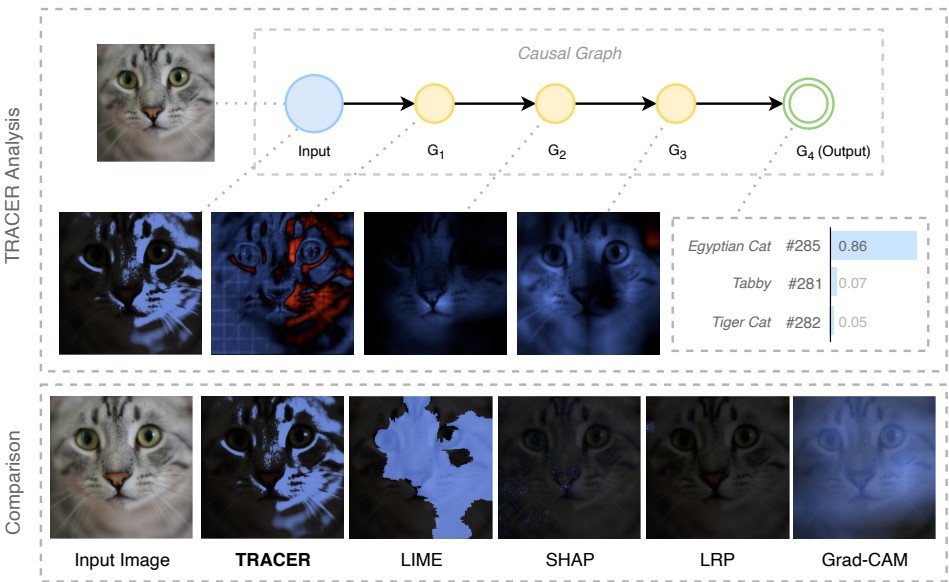

Figure 4: TRACER vs existing XAI methods using an ImageNet sample classified by ResNet-50. The second row shows feature contributions from different causal nodes, while the bottom row compares the explanations provided by different methods. The sparse explanations given by SHAP and LRP may require high-resolution screens for adequate visualization.

### 4.3 BEYOND LOCAL EXPLAINABILITY

To evaluate TRACER's capacity for global explainability, we integrated individual local explanations to form a comprehensive view of a model's decision logic. For this task, we focus on a random subset of the MNIST dataset, processed through the AlexNet architecture, to derive causal insights underpinning the classifier's decisions for all class samples. The results of this analysis, detailed in Appendix E, reveal significant redundancies within AlexNet's architecture for MNIST, allowing us to design compressed representations of the model to optimize the computational efficiency.

The characteristics and comparisons of these compressed models, reported in Table 1, show that the most refined model obtained (C1) exhibits a staggering 99.42% reduction in model size with only a 0.16% drop in accuracy. This highlights TRACER's potential for catalyzing practical innovations in DNN design and optimization, without undermining the predictive performance of these models.

Table 1: Comparison of TRACER-assisted compressed models. $\theta$ represents the number of parameters of the models, and Speed indicates the inference time per sample.

| Model | $\theta$ (M) | Size (MB) | FLOPs (M) | Speed (ms) | Accuracy (%) |
|---|---|---|---|---|---|
| AlexNet | 11.7 | 46.8 | 46.3 | $4.23^{\pm0.4}$ | **99.64** |
| C3 | 11.5 | 46.3 | 25.0 | $3.21^{\pm0.3}$ | **99.64** |
| C2 | 4.7 | 18.1 | 18.3 | $1.99^{\pm0.1}$ | 99.53 |
| C1 | **0.06** | **0.27** | **13.5** | $\mathbf{1.08^{\pm0.1}}$ | 99.48 |

### 4.4 DISCUSSIONS AND LIMITATIONS

We focused our evaluations of TRACER on accessible neural networks, where the internal architectures and intermediate activations can be directly analyzed. However, the flexibility and design of TRACER extend beyond these settings, making it equally applicable to black-box models where the internal dynamics remain obscured, and only the inputs and outputs are accessible. Under such constraints, TRACER remains valuable, offering two distinct avenues of exploration. First, it can analyze and quantify the influence of input features on the model's prediction. Alternatively, by using a surrogate model with an accessible structure, we can effectively approximate the underlying causal mechanisms driving the predictions. This adaptability underscores TRACER's potential in diverse environments.

While our TRACER approach is highly parallelizable by design, its depth of analysis can require a trade-off between granularity (the precision of the causal analysis determined by the number of interventions generated for each sample) and computational efficiency.

## 5 CONCLUSION

In this paper, we introduced TRACER, a novel approach for accurately estimating the causal dynamics embedded within deep neural networks. Through seamless integration of causal discovery and counterfactual analysis, our methodology enables a deep understanding of the decision-making processes of DNNs. Our empirical results demonstrate TRACER's ability to both identify the causal nodes and links underpinning a model's decisions, and also leverage counterfactuals to highlight the nuances that drive misclassifications, offering clear and actionable insights for model refinement and robustness. Beyond local explanations, we showcased the potential of our approach to capture the global dynamics of DNNs, leading to practical advantages such as novel and effective model compression strategies. Through our foundational principles and findings, we have ascertained that by producing intuitive, human-interpretable explanations, TRACER offers outstanding transparency to neural networks, significantly enhancing their trustworthiness for critical applications.

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

## A FRAMEWORK

### A.1 PROPOSITION 1 [CAUSAL ISOLATION OF INTERVENED SAMPLES]

*Proof (By induction).* Let $F : \mathcal{X} \to \mathcal{Y}$ be a DNN. For $x \in \mathcal{X}$, $I \subseteq \{1, \ldots, d\}$, and $b \in \mathbb{R}$, define the intervened sample $x'$ by:

$$x'_i = \begin{cases} b, & \text{if } i \in I; \\ x_i, & \text{if } i \notin I. \end{cases}$$

We will prove by induction on $n = |I|$ that $x'$ isolates the causal effect of features in $I$ on $F$.

**Base Case** ($n = 1$). Let $I = \{i\}$. Then $x'$ differs from $x$ only at index $i$:

$$x'_j = \begin{cases} b, & \text{if } j = i; \\ x_j, & \text{if } j \neq i. \end{cases}$$

Since only $x_i$ is altered, any change in $F(x')$ compared to $F(x)$ is due solely to the change in $x_i$. Thus,

$$F(x') - F(x) = \Delta F_i,$$

where $\Delta F_i$ represents the effect of changing $x_i$ to $b$.

**Inductive Step.** Assume the proposition holds for all subsets $I$ with $|I| = n$. Let $I' = I \cup \{i'\}$ with $|I'| = n + 1$ and $i' \notin I$. Define $x''$ by:

$$x''_i = \begin{cases} b, & \text{if } i \in I'; \\ x_i, & \text{if } i \notin I'. \end{cases}$$

Consider $F(x'')$. Since $x''$ differs from $x$ only at indices in $I'$, we have:

$$F(x'') - F(x) = (F(x'') - F(x')) + (F(x') - F(x)).$$

By the inductive hypothesis, $F(x') - F(x)$ isolates the effect of features in $I$, and $x''$ differs from $x'$ only at $i'$:

$$x''_i = \begin{cases} x'_i, & \text{if } i \neq i'; \\ b, & \text{if } i = i'. \end{cases}$$

Thus, the change from $F(x')$ to $F(x'')$ is due solely to $x_{i'}$:

$$F(x'') - F(x') = \Delta F_{i'},$$

where $\Delta F_{i'}$ represents the effect of changing $x_{i'}$ to $b$.

Combining the effects: $F(x'') - F(x) = \Delta F_{i'} + \sum_{i \in I} \Delta F_i$.

Therefore, $F(x'') - F(x)$ isolates the causal effects of features in $I' = I \cup \{i'\}$.

By induction on $n$, for any $I \subseteq \{1, \ldots, d\}$, the intervened sample $x'$ isolates the causal effect of features in $I$ on $F$. $\qquad\square$

### A.2 THEOREM 1 [LAYER GROUPING]

*Proof (By induction).* **Base case:** let us take two consecutive layers $f_i$ and $f_{i+1}$, such that $\mathcal{B}(K_{i+1}, K_i) = 1$. By the definition of the binary similarity matrix $\mathcal{B}$, $\mathcal{B}(K_{i+1}, K_i) = 1 \iff$ CKA$(K_{i+1}, K_i) \geq 1 - \epsilon$. This implies that the kernel representations $K_{i+1}$ and $K_i$ are sufficiently aligned:

$$K_{i+1} \approx K_i \quad \text{in terms of kernel alignment.}$$

Let us define a composite layer $g_{ii+1}(x) = f_{i+1} \circ f_i(x)$. Since CKA$(K_{g_{ii+1}}, K_{i+1}) = 1$, the composite layer $g_{ii+1}$ preserves the kernel similarity properties of $f_{i+1}$, satisfying $K_{g_{ii+1}} = K_{i+1}$. Thus, the two layers $f_i$ and $f_{i+1}$ can be replaced by $g_{ii+1}$ without altering the representation, establishing the base case.

**Induction step:** Let us assume that for $n$ layers $f_i, f_{i+1}, \ldots, f_{i+n-1}$, if $\mathcal{B}(K_{i+k}, K_{i+k-1}) = 1$ for all $k \in \{1, 2, \ldots, n - 1\}$, then these layers can be replaced by a composite layer $g_{i,i+n-1}(x) = f_{i+n-1} \circ \cdots \circ f_i(x)$, where $K_{g_{i,i+n-1}} = K_{i+n-1}$.

Let us now consider $n+1$ layers $f_i, f_{i+1}, \ldots, f_{i+n}$, with $\mathcal{B}(K_{i+k}, K_{i+k-1}) = 1$ for all $k \in \{1, 2, \ldots, n\}$. By the inductive hypothesis, the first $n$ layers $f_i, f_{i+1}, \ldots, f_{i+n-1}$ can be replaced by a composite layer $g_{i,i+n-1}(x)$, with $K_{g_{i,i+n-1}} = K_{i+n-1}$. Since $\mathcal{B}(K_{i+n}, K_{i+n-1}) = 1$, we know:

$$\text{CKA}(K_{i+n}, K_{i+n-1}) \geq 1 - \epsilon,$$

implying $K_{i+n} \approx K_{i+n-1}$. Let us define the new composite layer $g_{i,i+n}(x) = f_{i+n} \circ g_{i,i+n-1}(x)$. The kernel representation of $g_{i,i+n}$ satisfies:

$$K_{g_{i,i+n}} = K_{i+n}.$$

Therefore, the $n+1$ layers $f_i, f_{i+1}, \ldots, f_{i+n}$ can be replaced by the single composite layer $g_{i,i+n}$, preserving the representational similarity.

By the principle of mathematical induction, for any sequence of layers $\{f_i, f_{i+1}, \ldots, f_j\}$ such that $\mathcal{B}(K_k, K_{k-1}) = 1$ for all $k \in \{i+1, \ldots, j\}$, these layers can be replaced by a single composite layer $g_{ij}(x) = f_j \circ \cdots \circ f_i(x)$, with $K_{g_{ij}} = K_j$. Thus, $F'(x) = F(x)$, completing the proof. $\square$

## A.3 Theorem 2 [Necessary and Sufficient Conditions for Causal Nodes]

*Proof (By contradiction).* Let $F = f_k \circ f_{k-1} \circ \cdots \circ f_1$ be a DNN, where each $f_i : \mathbb{R}^{d_{i-1}} \to \mathbb{R}^{d_i}$, with $d_0 = n$ and $d_k = m$. And let $g = \{f_i\}_{i=r}^s$, $1 \leq r < s \leq k$, be a subset of $F$.

We will prove that $g$ is a causal node if and only if:

$$\forall i \in \{r, r+1, \ldots, s-1\}, \quad \text{CKA}(K_i, K_{i+1}) \geq 1 - \varepsilon,$$

where $K_i$ is the kernel (Gram) matrix of activations at layer $i$, and $\varepsilon \in (0, 1)$ is a threshold.

**Necessary Condition ($\Rightarrow$).** Let us assume $g$ is a causal node. And let us suppose, for contradiction, there exists $i \in \{r, \ldots, s-1\}$ such that: $\text{CKA}(K_i, K_{i+1}) < 1 - \varepsilon$.

This implies a significant dissimilarity between layers $i$ and $i+1$: $\|K_i - K_{i+1}\|_F^2 > \delta$, where $\delta$ corresponds to $\varepsilon$ and $\|\cdot\|_F$ denotes the Frobenius norm.

However, in a causal node, layers are by definition functionally redundant, so their activations should satisfy:

$$\|K_i - K_{i+1}\|_F^2 \leq \delta.$$

This contradiction implies: $\forall i \in \{r, \ldots, s-1\}, \quad \text{CKA}(K_i, K_{i+1}) \geq 1 - \varepsilon.$

**Sufficient Condition ($\Leftarrow$).** Let us assume: $\forall i \in \{r, \ldots, s-1\}, \quad \text{CKA}(K_i, K_{i+1}) \geq 1 - \varepsilon.$

And let us suppose, for contradiction, that $g$ is not a causal node. The composite function $g_{r,s}$ is given by: $g_{r,s} = f_s \circ f_{s-1} \circ \cdots \circ f_r$.

Since: $\text{CKA}(K_i, K_{i+1}) \geq 1 - \varepsilon, \quad \forall i \in \{r, \ldots, s-1\}$, the activations are highly similar:

$$K_i \approx K_{i+1}.$$

Therefore:

$$K_r \approx K_{r+1} \approx \cdots \approx K_s.$$

This implies that the composite function $g_{r,s}$ behaves similarly to a single layer $f_r$: $g_{r,s}(x) \approx f_r(x)$.

Thus, $g$ can be treated as a single causal node, contradicting the assumption that $g$ is not a causal node. Therefore, $g$ is a causal node. $\square$

## A.4 Minimization of Spurious Correlations

We will demonstrate that by using multiple interventions for our CKA analysis, TRACER minimizes the effects of spurious correlations in the identification of causal relationships within DNNs.

*Proof.* Let $F : \mathbb{R}^d \to \mathbb{R}^m$ be a deterministic DNN composed of $L$ layers: $F = f_L \circ f_{L-1} \circ \cdots \circ f_1$, where each $f_l$ represents the function of layer $l$. And let $x \in \mathbb{R}^d$ and $y = F(x) \in \mathbb{R}^m$ be an input vector and its corresponding output by the network.

Let us consider a set of $N$ interventions, resulting in inputs $\{x^{(i)}\}_{i=1}^N$, where each $x^{(i)}$ is obtained by modifying a subset of features in $x$: $x^{(i)} = x + \delta^{(i)}$, with $\delta^{(i)} \in \mathbb{R}^d$ being the intervention vector for the $i$-th intervention, where $\delta_j^{(i)} \neq 0$ only for features being intervened upon.

For each input $x^{(i)}$, the activations at layers $l$ and $m$ are computed as follows:

$$h_l^{(i)} = f_l \circ f_{l-1} \circ \cdots \circ f_1(x^{(i)}),$$

$$h_m^{(i)} = f_m \circ f_{m-1} \circ \cdots \circ f_{l+1}(h_l^{(i)}).$$

Let $A_l$ and $A_m$ be the matrices collecting the activations across interventions:

$$A_l = \begin{bmatrix} (h_l^{(1)})^\top \\ (h_l^{(2)})^\top \\ \vdots \\ (h_l^{(N)})^\top \end{bmatrix} \in \mathbb{R}^{N \times d_l}, \quad A_m = \begin{bmatrix} (h_m^{(1)})^\top \\ (h_m^{(2)})^\top \\ \vdots \\ (h_m^{(N)})^\top \end{bmatrix} \in \mathbb{R}^{N \times d_m}.$$

The linear kernel matrices for CKA can then be computed as: $K_{A_l} = A_l A_l^\top \in \mathbb{R}^{N \times N}$, $K_{A_m} = A_m A_m^\top \in \mathbb{R}^{N \times N}$, with centered kernel matrices: $\tilde{K}_{A_l} = H K_{A_l} H$, $\tilde{K}_{A_m} = H K_{A_m} H$, where $H = I_N - \frac{1}{N} \mathbf{1}_N \mathbf{1}_N^\top$ is the centering matrix, $I_N$ is the $N \times N$ identity matrix, and $\mathbf{1}_N$ is an $N \times 1$ vector of ones.

Given these kernel matrices, the Hilbert-Schmidt Independence Criterion (HSIC) and the CKA can be obtained:

$$\text{HSIC}(A_l, A_m) = \text{Tr}(\tilde{K}_{A_l} \tilde{K}_{A_m}),$$

$$\text{CKA}(A_l, A_m) = \frac{\text{HSIC}(A_l, A_m)}{\sqrt{\text{HSIC}(A_l, A_l)\,\text{HSIC}(A_m, A_m)}}.$$

**True Causal Relationship.** Let us assume a true causal relationship between layers $l$ and $m$, so $h_m^{(i)}$ is a deterministic function of $h_l^{(i)}$: $h_m^{(i)} = f_{m:l+1}(h_l^{(i)})$, where $f_{m:l+1} = f_m \circ f_{m-1} \circ \cdots \circ f_{l+1}$.

Under multiple interventions, variations in $h_l^{(i)}$ lead to corresponding variations in $h_m^{(i)}$, preserving their functional relationship. Hence, the covariance between $h_l^{(i)}$ and $h_m^{(i)}$ remains high. This cross-covariance matrix is given by:

$$C_{lm} = \frac{1}{N} \bar{A}_l^\top \bar{A}_m \in \mathbb{R}^{d_l \times d_m},$$

where $\bar{A}_l$ and $\bar{A}_m$ are the centered activation matrices:

$$\bar{A}_l = A_l - \frac{1}{N} \mathbf{1}_N \mathbf{1}_N^\top A_l, \quad \bar{A}_m = A_m - \frac{1}{N} \mathbf{1}_N \mathbf{1}_N^\top A_m.$$

Similarly, we can obtain the auto-covariance matrices as:

$$C_{ll} = \frac{1}{N} \bar{A}_l^\top \bar{A}_l, \quad C_{mm} = \frac{1}{N} \bar{A}_m^\top \bar{A}_m,$$

along with their Frobenius norms:

$$\|C_{lm}\|_F^2 = \sum_{i=1}^{d_l} \sum_{j=1}^{d_m} (C_{lm})_{ij}^2, \quad \|C_{ll}\|_F^2 = \sum_{i=1}^{d_l} \sum_{j=1}^{d_l} (C_{ll})_{ij}^2, \quad \|C_{mm}\|_F^2 = \sum_{i=1}^{d_m} \sum_{j=1}^{d_m} (C_{mm})_{ij}^2.$$

Since $h_m^{(i)}$ is a function of $h_l^{(i)}$, the covariance $C_{lm}$ is significant, so $\|C_{lm}\|_F^2$ is large. Therefore, the CKA similarity is high:

$$\text{CKA}(A_l, A_m) = \frac{\|C_{lm}\|_F^2}{\sqrt{\|C_{ll}\|_F^2 \|C_{mm}\|_F^2}} \approx 1.$$

**Spurious Correlation.** For spurious correlations, there is no functional dependence between $h_l^{(i)}$ and $h_m^{(i)}$. Therefore, the cross-covariance $C_{lm}$ is small, so $\|C_{lm}\|_F^2$ is negligible. Thus, the CKA similarity is low:

$$\mathrm{CKA}(A_l, A_m) = \frac{\|C_{lm}\|_F^2}{\sqrt{\|C_{ll}\|_F^2 \|C_{mm}\|_F^2}} \approx 0.$$

By setting a threshold $\tau = 1 - \epsilon$ close to 1, we distinguish between true causal connections and spurious correlations:

$$\mathrm{CKA}(A_l, A_m) \geq \tau \implies \text{Causal Connection,}$$
$$\mathrm{CKA}(A_l, A_m) < \tau \implies \text{Spurious Correlation.}$$

$\square$

## B Feature Attributions at Causal Nodes

Figure 5 below depicts how individual features contribute to the network's final decision. For every causal node (group of neural network layers), we highlight the top contributing features (top convolution filter output or top-3 feature outputs for linear layers). Positive contributions are distinctly marked in blue, signifying features that positively influence the network's decision, while negative contributions are depicted in red, pointing out the features that negatively affect the decision.

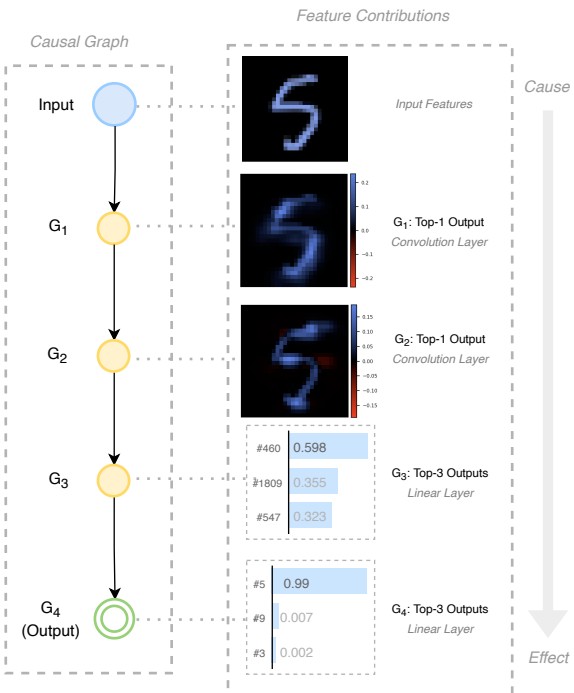

Figure 5: Contribution of features within each causal node. Blue and red respectively indicates positive and negative contributions. The overlay on the input sample provides a cohesive visualization of how distinct features of the input affect the final decision via the causal mechanism discovered.

## C Counterfactual Analysis

The objective of counterfactual generation in the context of our research is to offer interpretable insights into the decision-making processes of deep neural networks, particularly in cases of misclassification. By examining the contrast between the original input and the generated counterfactual, we can uncover subtle features or patterns that influence the model's decision, thereby pinpointing what changes might rectify misclassifications.

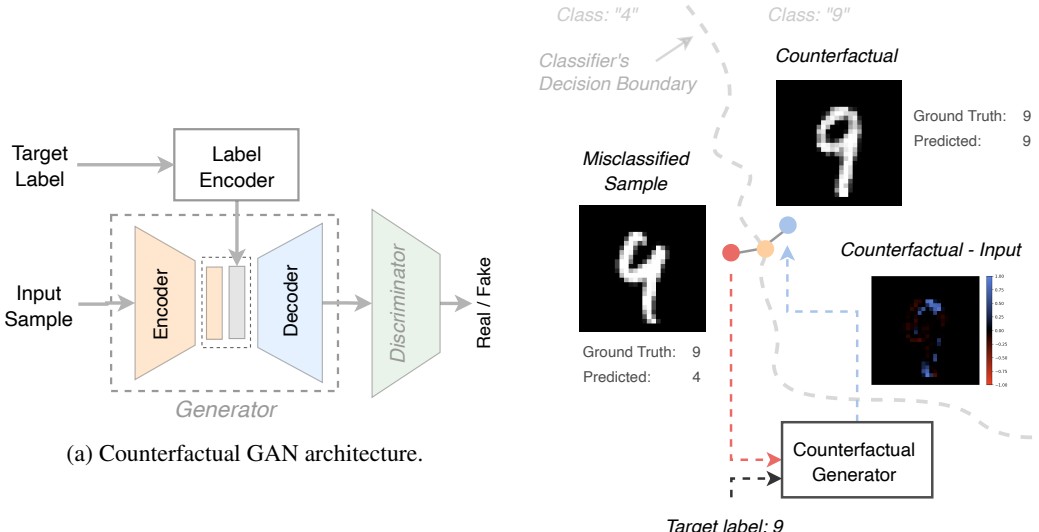

(a) Counterfactual GAN architecture.

(b) Illustration of a misclassified MNIST sample and its generated counterfactual.

Figure 6: Architecture of the counterfactual generator and illustration of its generation process.

To this end, we design a GAN architecture tailored to our counterfactual generation tasks. This GAN, depicted in Figure 6a, consists of a CNN-based Generator for creating plausible, class-conditional counterfactuals, coupled with a CNN-based Discriminator analyzing the authenticity of the generated images, is designed as follows:

1. The encoder uses convolutional layers to transform the inputs into latent embeddings which are then merged with class information via label encodings;

2. The decoder uses transposed convolutions to construct the counterfactual inputs from the augmented latent representations produced by the encoder.

In this experiment, we explore the use of counterfactuals as a means to understand causes for misclassifications, as well as identify the minimal feature changes required to obtain correct outcomes. Figure 7 below shows examples of counterfactuals generated for handwritten digits (MNIST) and natural images (CIFAR-10) (Krizhevsky et al., 2009).

As illustrated in Figure 6b, given an initially misclassified input and a desired target label, our GAN-based counterfactual generator produces an alternative version of the input, which, when fed to the model, results in the desired outcome. The differences between the input and its counterfactual reveal the minimal modifications required for the classifier to produce the correct (desired) decision.

Through a side-by-side analysis of the causal mechanisms obtained from the predictions of the misclassified sample and its counterfactual, TRACER provides clear insights into the primary contributing factors to the initial misclassification, while also highlighting via the counterfactual's analysis, the optimal neural pathways for the network to yield the correct (and desired) outcome. This detailed causal analysis is visually represented in Figure 8. Upon examination, we discern that a predominant portion of the input features, represented in blue, activate neurons that steer the classifier towards the produced outcome in both cases. However, the causal analysis of the misclassified sample reveals a notably more extensive set of features that oppose the predicted outcome when contrasted with the counterfactual. This observation makes it evident that TRACER not only identifies which parts of the input features support the misclassification (in blue) but also which features contradict this decision (in red). Intriguingly, while the causal graphs remain consistent for both inputs in this instance, the classifier's activations manifest pronounced differences. This insight suggests that the model's learned parameters might lack the flexibility to generalize enough to correctly discern the true label of the misclassified sample. To address this, potential avenues might include incorporating such

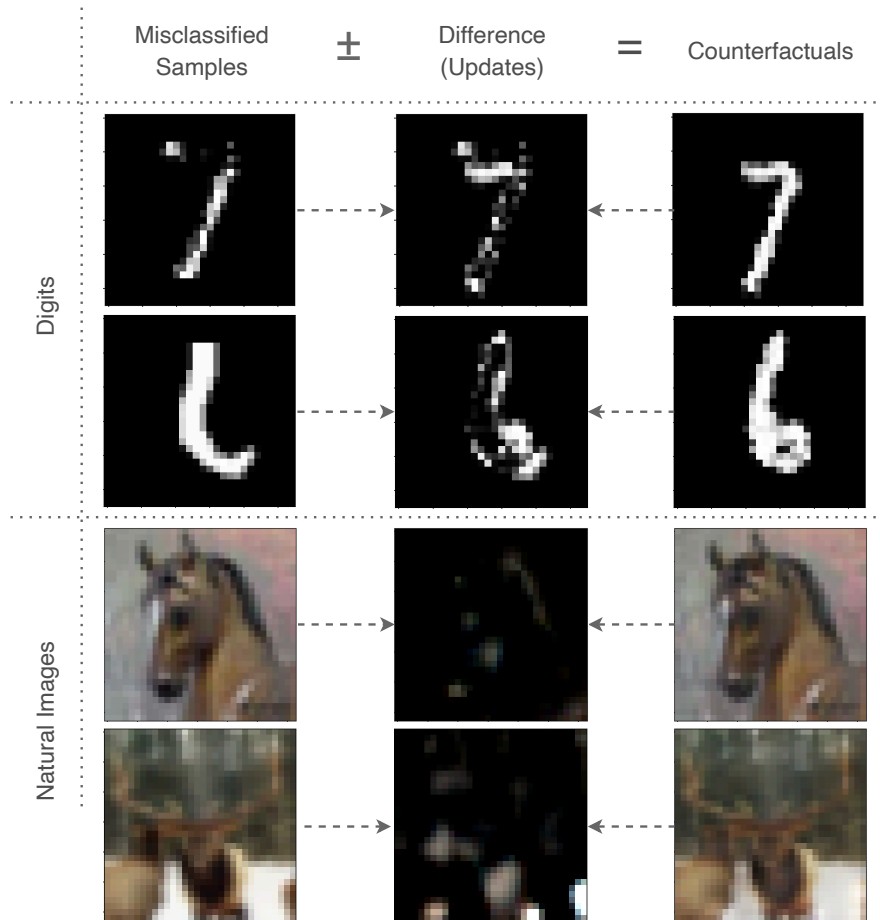

Figure 7: Illustration of counterfactuals generated for handwritten digits and natural images.

misclassified instances into the training set or fine-tuning the model with regularization techniques to enhance its generalization capabilities. In essence, counterfactuals provide a means to understand model decisions and identify changes to input features that can correct misclassifications.

This causal analysis reveals that our counterfactual generation method serves two main purposes. First, it provides an intuitive visualization for understanding the nuances of model decisions. Secondly, from a model development and refinement perspective, these counterfactuals can highlight potential vulnerabilities or biases in the model, guiding further training or fine-tuning endeavours.

## D  GENERALIZATION

### D.1  IMAGE DATASETS

Here, we address the question of scalability of TRACER to large-scale image datasets. Given the challenges associated with the explainability of real-world images (e.g., the intricacies of pixel-level interactions, variances in image quality, or scale), we use for this task the MNIST and ImageNet (Deng et al., 2009) datasets, classified with the AlexNet and ResNet-50 architectures respectively. Using the ImageNet dataset, known for its vastness, diversity, and complexity, we show that TRACER overcomes the limitations of existing explainability methods. The explanations produced by TRACER and benchmark explainability methods are depicted in Figure 9, showing that while existing methods

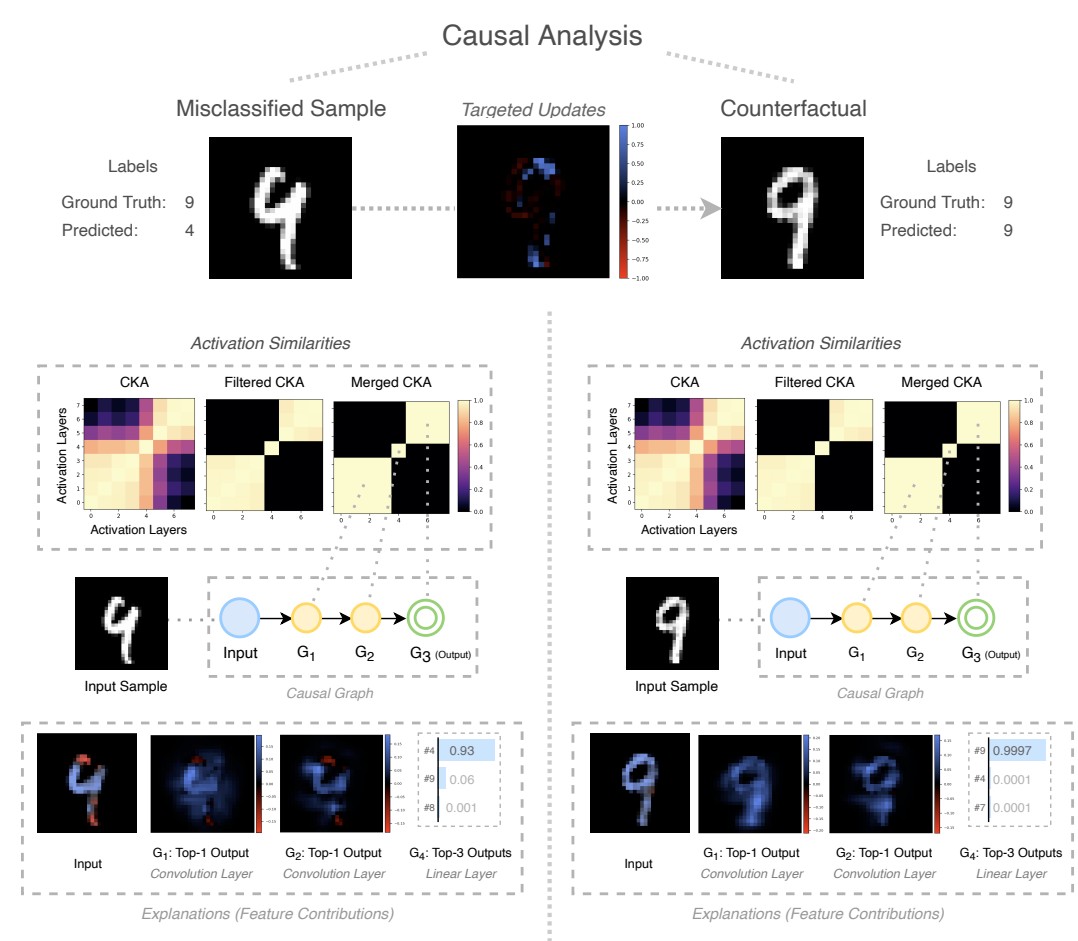

Figure 8: Comparison of an original misclassified input, the generated counterfactual, and the associated causal mechanisms. The variations between the original and counterfactual inputs highlight the pertinent features influencing the model's decision-making process. (Blue: Positive contributions; Red: Negative contributions; $G_i$: $i$-th Layer Group)

struggle to produce coherent and comprehensive explanations, TRACER consistently reveals the core features and patterns crucial for classification decisions. The effectiveness of our proposed approach becomes even more apparent when used with complex models like ResNet-50, as it still maintains its precision despite the intricate patterns leveraged by very deep networks, emphasizing its capability to accurately discern the nuances of complex interactions within deeper architectures.

In contrast to TRACER,

- Every execution of *LIME* produces different explanations due to its inherent stochasticity, hindering interpretability.

- *SHAP* and *LRP* explanations produce misleading results due to their sensitivity to model and dataset complexities, resulting in overly detailed or sparse attributions that do not always intuitively align with the underlying data patterns.

- As *Grad-CAM* explanations are based on the coarse spatial resolution of the final convolutional layer of a DNN, this method often leads to highlighting broader regions rather than precise feature-level contributions to the decision-making.

- *LRP* and *Grad-CAM*, inherently designed for white-box DNNs, where internal model structures are accessible, face significant restrictions in terms of applicability and utility in scenarios involving black-box or proprietary models.

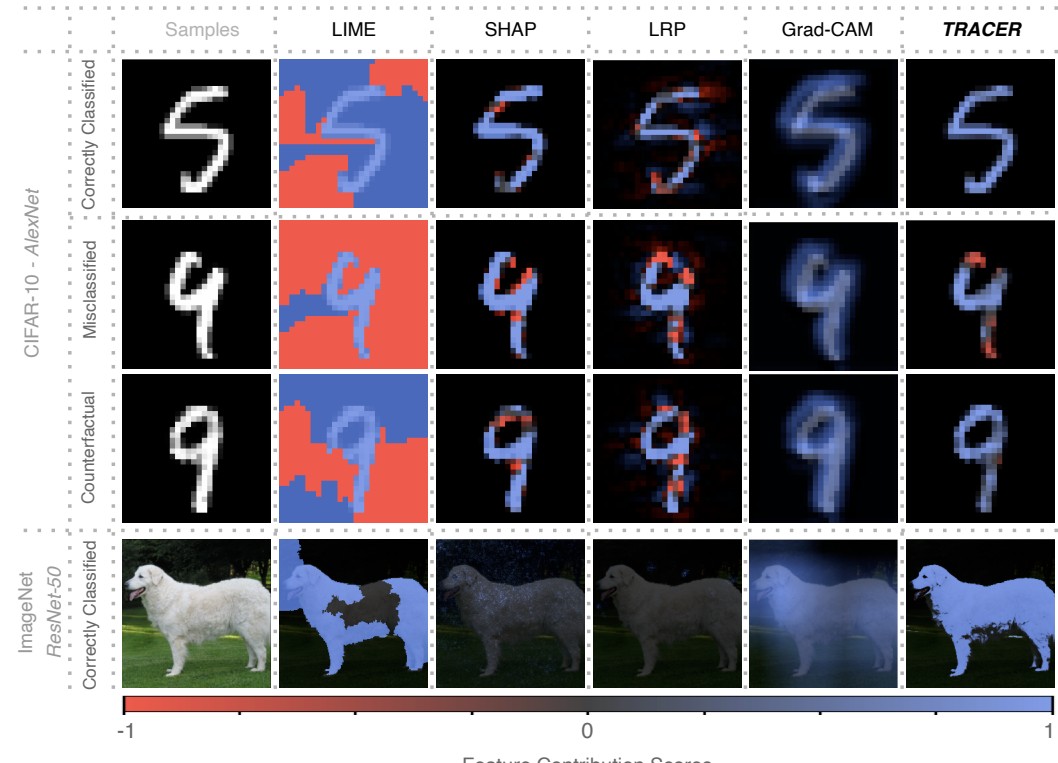

Figure 9: Comparison of TRACER results against existing explainability methods.

## D.2 TABULAR DATASETS

Transitioning from the realm of images, we further explored the efficacy of TRACER in the context of structured (or tabular) data. For this endeavour, we selected the CIC-IDS 2017 (Sharafaldin et al., 2018) network traffic dataset, which is representative of real-world network behaviors and patterns. This dataset poses its own set of challenges, distinct from image datasets, such as the mix of categorical and numerical attributes, the potential correlations between features, and the variance in feature scales.

The results presented in Figure 10 illustrate TRACER's ability to provide detailed and accurate explanations beyond the image domain. For the sample explained in this figure, where a network traffic generated during a DDoS attack is considered as benign traffic by a multi-layer feed-forward neural network classifier, we observe that the features indicative of an attack negatively contribute to the decision of the classifier. Specifically, the explanations provided tell us which features were found relevant for classifying this network traffic as an attack (i.e., Source/Destination Port numbers, frequency of communication, sizes of transferred data, etc.).

The clarity of the causal explanations obtained by TRACER for such tasks make it particularly suitable given the criticality of network intrusion detection systems in ensuring cybersecurity, where the ability to transparently understand and trust decisions can be indispensable for the practical viability of such systems.

## E GLOBAL EXPLAINABILITY

Given the effectiveness of TRACER in explaining neural network decisions for individual samples, we endeavour to evaluate its potential as a global explainability tool to paint a holistic picture of the model's decision-making. To this end, rather than solely relying on global explanations, which might overlook individual nuances, we adopt an approach that aggregates local explanations to derive a global perspective. Specifically, using TRACER, we perform local explanations on a

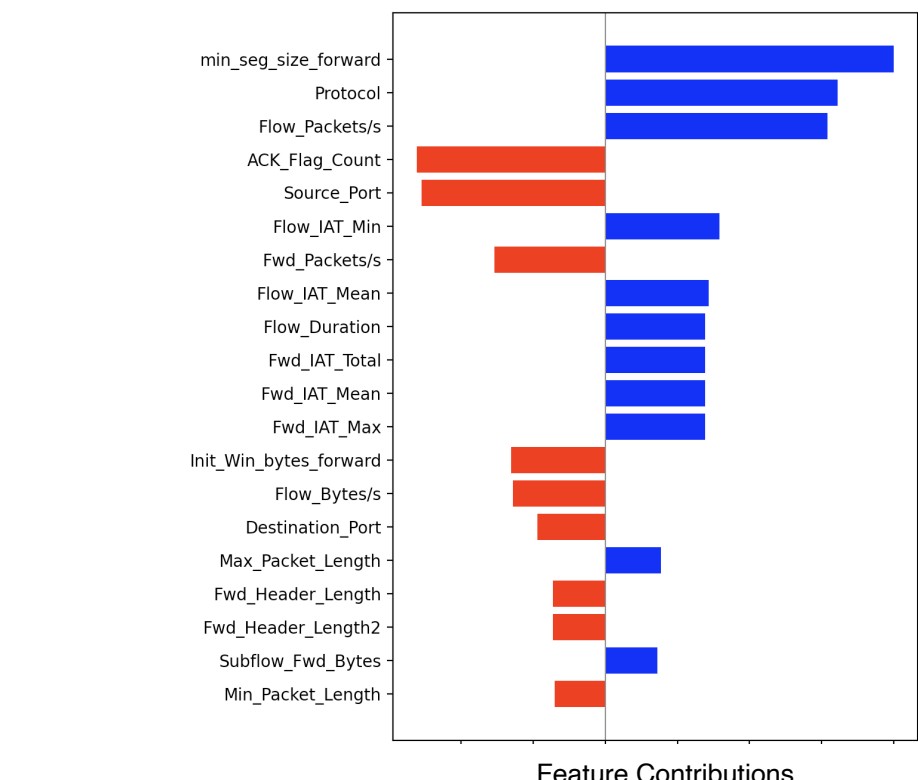

Figure 10: Explainability of tabular datasets with TRACER. A sample from a Network Intrusion Detection dataset is misclassified as benign traffic rather than its correct class (DDoS attack). Negative contributions are shown in red and positive contributions in blue for the top-20 features.

strategically selected subset of the dataset, aiming to capture a representative understanding of the overall characteristics. For this experiment, we selected the MNIST dataset classified using the AlexNet architecture as before. While without loss of generality, simply performing random sampling within all classes suffices for this experiment, by using clustering algorithms (Settles, 2009; Olvera-López et al., 2010) or Proximally-Connected graphs (Diallo & Patras, 2023), more optimal sampling policies can also be adopted to identify and select the most influential samples. Our findings for this experiment revealed several remarkable insights into the potential of TRACER, and into the use of AlexNet for MNIST classification.

Specifically, as shown in Figure 11:

1. About 85% of the samples could be concisely explained with a causal mechanism entailing merely 2 intermediate causal nodes. This level of generalization showcases the simplicity of the model's decision-making processes.
2. With just one additional causal node, the causal mechanism explains 99% of the classifications, bringing the total to 3 intermediate causal nodes.
3. To attain a full coverage, explaining 100% of the classifications, the complexity increases only marginally, requiring 4 intermediate causal nodes.

Encouraged by these insights into the causal dynamics of AlexNet's decisions on the MNIST dataset, we venture to create compressed representations of the original model. The objective is twofold: preserving the original model's accuracy while substantially reducing its computational complexity. Leveraging the knowledge distilled from TRACER, we craft the corresponding compressed models by replacing redundant layers in a layer group with a single representative layer where inputs and output sizes are adjusted to work with the rest of the network. The compressed models C1, C2, and C3, respectively corresponding to initial coverages of C1: 84.6%, C2: 98.8%, and C3: 100%, are then trained on the identical training set as the original model. The results, presented in Table 1, show that the most compressed model achieves a staggering 99.42% reduction in model size,

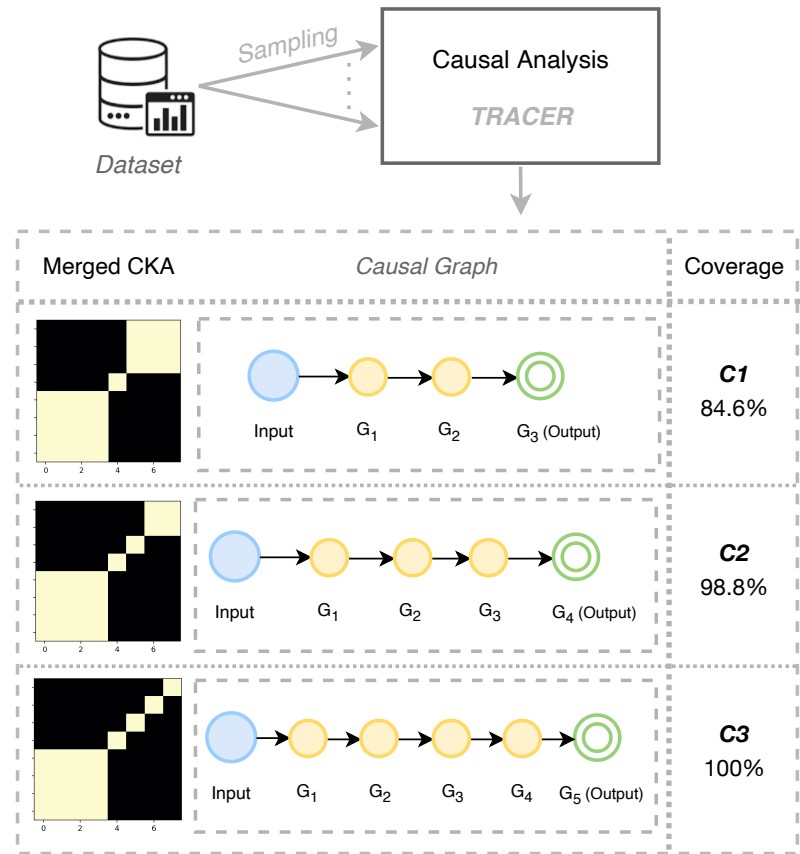

Figure 11: Global explainability with TRACER– Generalization of causal mechanisms across samples. The Coverage column indicates the percentage of analyzed samples that can be explained by distinct causal mechanisms.

while only sacrificing a negligible 0.16% in accuracy, making it significantly more lightweight and computationally efficient.

By decoding the fundamental causal interactions within neural networks, this experiment shows that TRACER's capacity to provide global explanations and insights can also inspire practical applications such as model compression, without compromising the integrity of the predictions. Furthermore, it is worth noting that the compressed models derived through our approach remain fully compatible with existing and well-established compression methods such as quantization and pruning, further extending their efficiency and applicability across diverse deployment scenarios.

We provide a comprehensive comparison between the baseline MNIST architecture and its three compressed versions (C1, C2, and C3) obtained with TRACER. Table 2 below includes detailed information about the layers in each model (such as layer type, output dimensions, and the number of parameters), as well as classification accuracies and compression ratios.

Table 2: Comparison between the baseline MNIST architecture and its compressed versions. For each version, the layer configurations, output shapes, classification accuracies, and parameter counts are presented, highlighting how TRACER-enabled compressed models preserve performance.

| Layer (Type) | Baseline | C3 | C2 | C1 |
|---|---|---|---|---|
| Conv2d | $32 \times 26 \times 26$ | - | - | - |
| ReLU | $32 \times 26 \times 26$ | - | - | - |
| Conv2d | $64 \times 26 \times 26$ | $64 \times 26 \times 26$ | $64 \times 26 \times 26$ | $64 \times 26 \times 26$ |
| ReLU | $64 \times 26 \times 26$ | $64 \times 26 \times 26$ | $64 \times 26 \times 26$ | $64 \times 26 \times 26$ |
| MaxPool2d | $64 \times 13 \times 13$ | - | - | - |
| Conv2d | $96 \times 13 \times 13$ | - | - | - |
| ReLU | $96 \times 13 \times 13$ | - | - | - |
| Conv2d | $64 \times 13 \times 13$ | - | - | - |
| ReLU | $64 \times 13 \times 13$ | - | - | - |
| Conv2d | $32 \times 13 \times 13$ | $32 \times 26 \times 26$ | $32 \times 26 \times 26$ | $32 \times 26 \times 26$ |
| ReLU | $32 \times 13 \times 13$ | $32 \times 26 \times 26$ | $32 \times 26 \times 26$ | $32 \times 26 \times 26$ |
| MaxPool2d | $32 \times 12 \times 12$ | $32 \times 12 \times 12$ | $32 \times 12 \times 12$ | $32 \times 12 \times 12$ |
| Dropout | 4608 | 4608 | 4608 | 4608 |
| Linear | 2048 | 2048 | - | - |
| ReLU | 2048 | 2048 | - | - |
| Dropout | 2048 | 2048 | - | - |
| Linear | 1024 | 1024 | 1024 | - |
| ReLU | 1024 | 1024 | 1024 | - |
| Linear | 10 | 10 | 10 | 10 |
| **Accuracy (%)** | 99.64 | 99.64 | 99.53 | 99.48 |
| **Total Parameters** | 11,696,202 | 11,567,786 | 4,749,994 | 66,218 |
| **Compression (%)** | - | 1.1% | 59.4% | 99.4% |

## F  IMPLEMENTATION DETAILS

**Counterfactual Generator.** The counterfactual GAN architecture was designed with the following hyperparameters:

- Learning rate: $10^{-3}$
- Balancing coefficient ($\lambda$): 0.1
- Number of training epochs: 100
- Regularization metric: $\ell_1$-norm

The generator consists of four convolutional layers in the encoder, combined with class information via one-hot encoding, and a decoder with transposed convolutions to produce counterfactual instances.

**Intervention Setup.** We design interventions to occlude portions of the inputs before normalization:

- For MNIST: 3x3 patches with a 1x1 sliding window
- For ImageNet: 30x30 patches with a 15x15 sliding window
- For CIC-IDS 2017: 1x3 patches with a 1x1 sliding window

**Training and Optimization Details.** TRACER is implemented in PyTorch (Paszke et al., 2019), and all DNN models are trained using the Adam optimizer with a learning rate of $10^{-3}$. Unless stated otherwise, default parameters are used throughout all training processes.

**CKA Threshold for Layer Grouping.** For causal analysis, the Centered Kernel Alignment (CKA) similarity threshold was set to $1 - \epsilon$, where $\epsilon$ represents a small tolerance for dissimilarity. In our experiments, $\epsilon$ was set to 0.05.

# G IMPACT OF CKA SENSITIVITY ON CAUSAL STRUCTURES

In this section, we investigate the sensitivity of TRACER's causal discovery process to the choice of the threshold parameter $\epsilon$ used in the CKA similarity measure for grouping layers into causal nodes. This threshold determines the margin of dissimilarity allowed between layers to be considered functionally similar and thus grouped into a single causal node. We conduct ablation studies on both the MNIST and ImageNet datasets to understand how varying $\epsilon$ influences the structure and interpretability of the discovered causal graphs.

For this analysis, we vary $\epsilon$ across a range of values $(0, 0.05, 0.1, 0.2, 1)$ and observe the resulting causal graph structures. Lower $\epsilon$ implies stricter similarity criteria, leading to more granular groupings, whereas a higher $\epsilon$ allows for broader groupings by tolerating greater dissimilarity between layers.

We apply these varying thresholds to the trained MNIST and ImageNet models, performing for each value of $\epsilon$ the following steps:

1. Compute the CKA similarity between consecutive layers.

2. Group layers into causal nodes based on the CKA threshold.

3. Construct the causal graph with the identified causal nodes and their interconnections.

## G.1 RESULTS AND ANALYSIS

### G.1.1 MNIST CLASSIFICATION

Figure 12 illustrates the causal graphs obtained for different $\epsilon$ values when applied to the MNIST classifier.

**Observations:**

- $\epsilon = 0$: The stringent threshold results in highly granular groupings, treating individual layers as separate causal nodes. This leads to a complex causal graph with numerous nodes and connections, hindering interpretability.

- $\epsilon = 0.05$: A moderate threshold beginning to merge some consecutive layers into single causal nodes. Key functional blocks of the network are now represented as single causal nodes, reducing the complexity of the causal graph while maintaining meaningful distinctions between different processing stages, thereby enhancing interpretability without substantial loss of detailed information.

- $\epsilon = 0.1$: Further increasing $\epsilon$ results in broader groupings, significantly simplifying the causal graph.

- $\epsilon = 0.2$: At these higher thresholds, the causal graph becomes increasingly abstract, with major sections of the network consolidated into larger causal nodes. While this simplifies the graph, it may oversimplify the underlying causal mechanisms, potentially obscuring finer-grained interactions.

- $\epsilon = 1$: The threshold becomes non-restrictive, allowing all layers to be grouped into a single causal node. This results in a highly abstract causal graph with a single node representing the entire network, eliminating any internal structural distinctions. While the graph is exceedingly simple, it offers no meaningful insights into the network's internal decision-making processes, effectively rendering the causal discovery uninformative.

### G.1.2 IMAGENET CLASSIFICATION

Similarly, Figure 13 presents the causal graphs derived from the ResNet-50 model trained on ImageNet for varying $\epsilon$ values. For this experiment, we select a sample that shows the effects of residual connections on the classifier's behaviour.

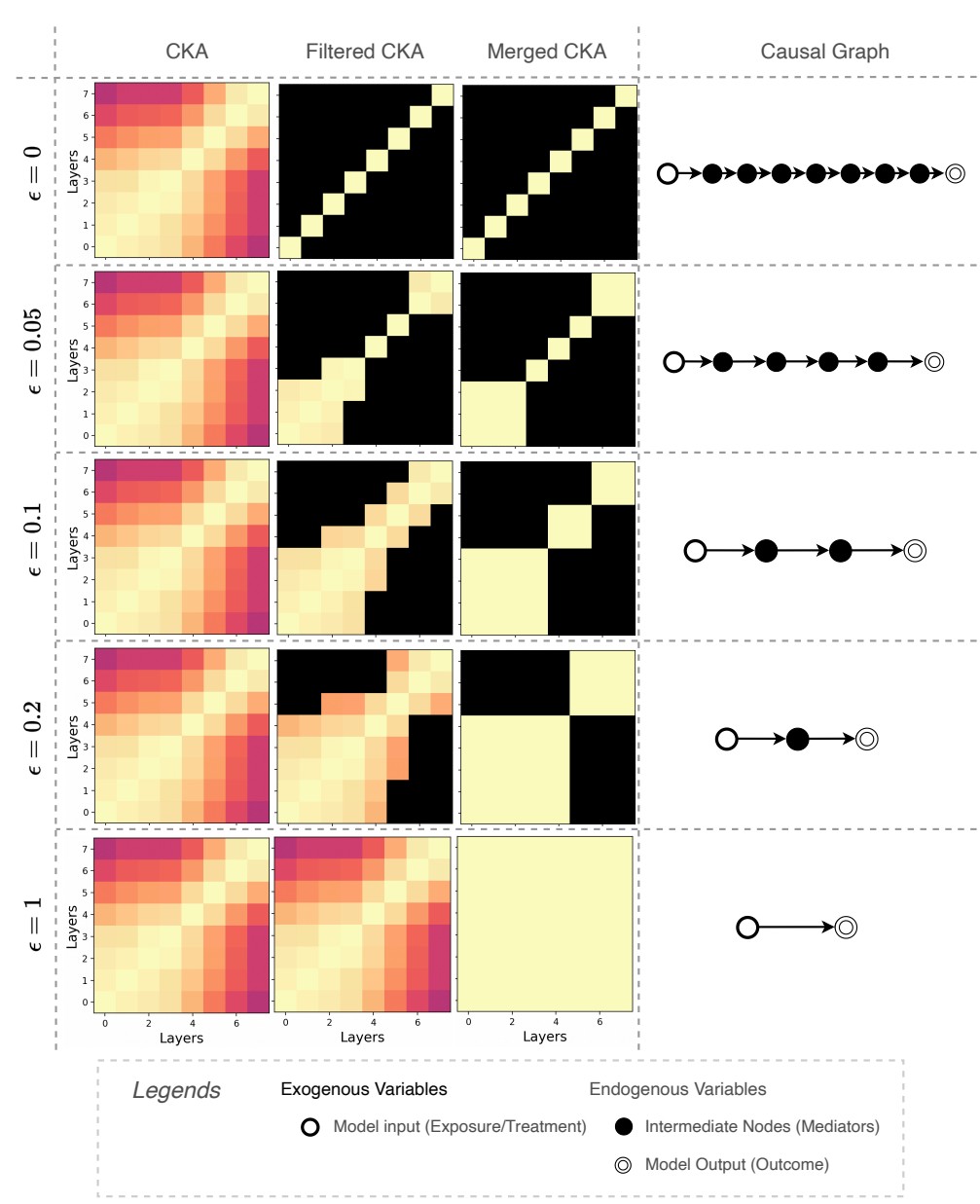

Figure 12: Impact of $\epsilon$ on the causal graphs discovered by TRACER when processing a given sample with the MNIST classifier. Each row corresponds to a different $\epsilon$ value, demonstrating how the granularity of layer groupings evolves.

**Observations:**

- $\epsilon = 0$: The high-resolution causal graph reflects the intricate architecture of ResNet-50, with each block treated as a functionality distinct layer. While accurate, the resulting graph is highly complex.

- $\epsilon = 0.05$: The threshold begins to merge similar residual blocks into single causal nodes, capturing the repetitive nature of ResNet architectures and reducing the complexity of the causal graph. This abstract view of ResNet-50 groups functionally similar blocks as single

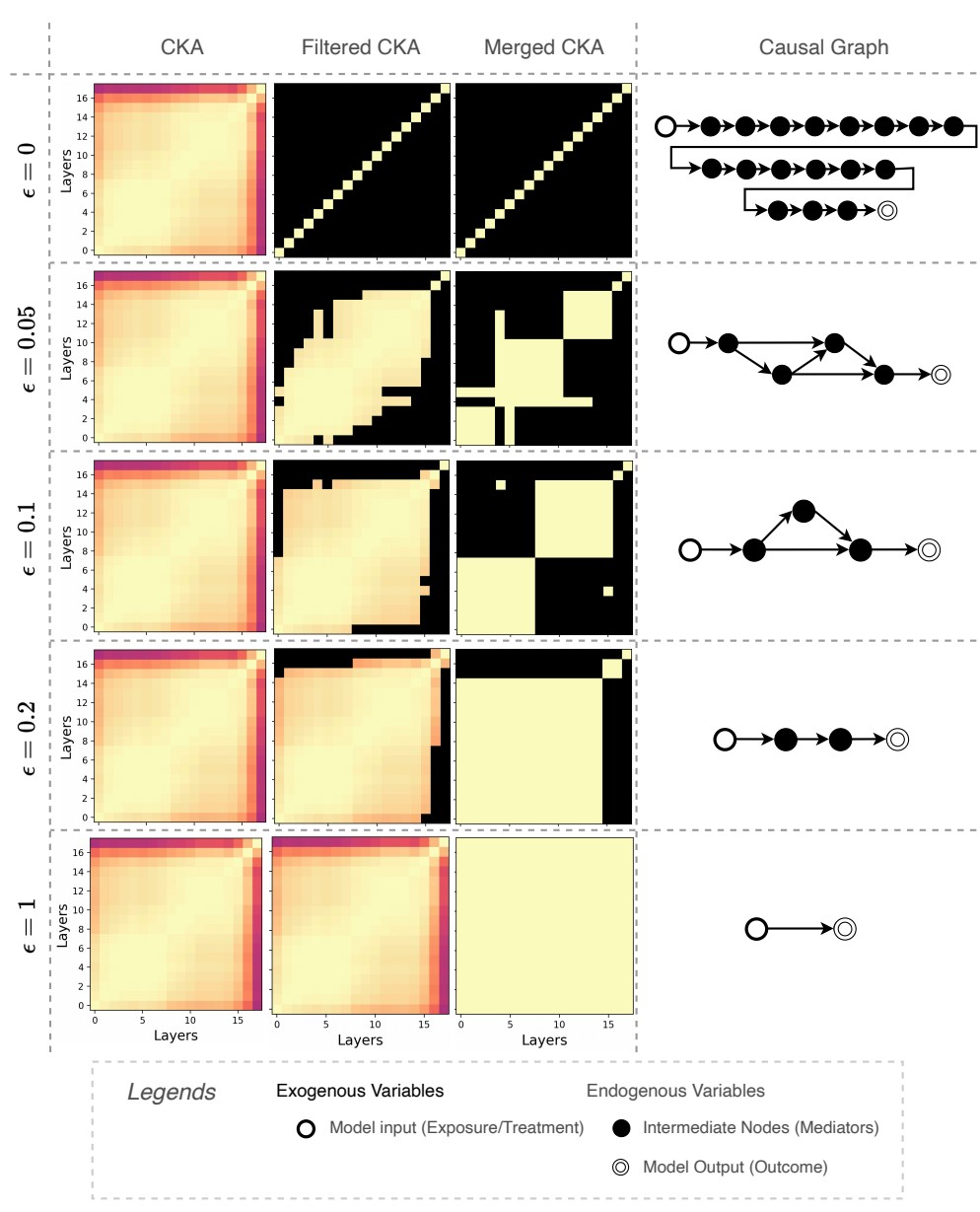

Figure 13: Impact of $\epsilon$ on the causal graphs discovered by TRACER when processing a given sample with the ImageNet classifier. Each row corresponds to a different $\epsilon$ value, demonstrating how the granularity of layer groupings evolves.

causal nodes, while showing the effects of residual connections, thereby striking a balance between structural detail and interpretability.

- $\epsilon = 0.1$: A noticeable simplification occurs, with entire stages of ResNet-50 being represented as single causal nodes.

- $\epsilon = 0.2$: The causal graph becomes highly abstract, with large sections of the network encapsulated within single nodes. While this enhances overall interpretability, it overlooks important nuances in layer interactions critical for understanding the model's decision-making process.

- $\epsilon = 1$: The non-restrictive threshold groups all layers into a single causal node, resulting in an overly abstract causal graph with just one node representing the entire ResNet-50 architecture. While the graph is exceedingly simple and easy to interpret, it offers no meaningful insights into the network's internal decision-making processes or the hierarchical structure of residual blocks. Consequently, the causal discovery becomes uninformative, as it fails to capture the distinct functional roles of different network segments.

### G.2    Trade-offs and Recommendations

The ablation studies reveal a clear trade-off between the granularity of causal graph representations and their interpretability:

- **Lower values (e.g., $\epsilon = 0$):** Offers a detailed view of the network's internal mechanisms but results in complex and potentially cumbersome causal graphs.
- **Moderate values (e.g., $\epsilon \in [0.05, 0.2]$):** Balances detail and simplicity, providing clear and interpretable causal structures while retaining meaningful distinctions between different network components.
- **Higher values (e.g., $\epsilon > 0.2$):** Simplifies the causal graph significantly but may obscure important layer interactions and reduce the depth of insights obtainable from the causal analysis.

Based on these findings, an $\epsilon$ value in the range of 0.05 to 0.2 is optimal for achieving a balance between interpretability and detail, as this allows TRACER to produce causal graphs that are abstract while being sufficiently detailed to provide meaningful insights into the network's behaviour.

## H    Causal Discovery for Latent-Space-Based Models

In this section, we will assess TRACER's suitability for studying the causal structures of more complex models, such as U-Nets Ronneberger et al. (2015) and Variational Autoencoders (VAEs) Kingma (2013).

### H.1    Spatially-Structured Models

Models like U-Net are renowned for their ability to capture both local and global features through skip connections between their downsampling and upsampling paths. Such spatially-structured models are widely used in image segmentation and classification tasks but pose significant challenges for interpretability due to their complexity. To evaluate the ability of TRACER to handle such architectures, we apply our causal discovery method to a U-Net model trained on the CIFAR-10 dataset. The architecture consists of four down-sampling blocks and by four up-sampling blocks, followed by a classification layer, with each block containing convolutional layers and non-linear activations, designed to progressively encode and decode spatial information.

As illustrated in Figure 14, our causal discovery process resulted in a causal graph with five nodes representing the key stages of the network: two causal nodes for the down-sampling path, two causal nodes for the up-sampling path, and one causal node for the classification layer. With non-adjacent causal links discovered between the first down-sampling causal node and the last up-sampling causal node, as well as between the last down-sampling and first-up-sampling causal nodes (corresponding to the skip connections inherent in the U-Net architecture), TRACER produces an accurate high-level causal view of the network, providing insights into how different parts of the network interact and contribute to the classification node, thereby improving the interpretability of the network.

### H.2    Stochastic Models

Inherently stochastic models, such as Variational Autoencoders (VAEs), introduce randomness into a network's operations through stochastic layers, which can present challenges for causal analysis due to variables that are not directly observed and may not be deterministic. To address this, we explicitly account for stochastic components by modelling stochastic computations as separate layers and treating stochasticity as exogenous variables in the causal graph.

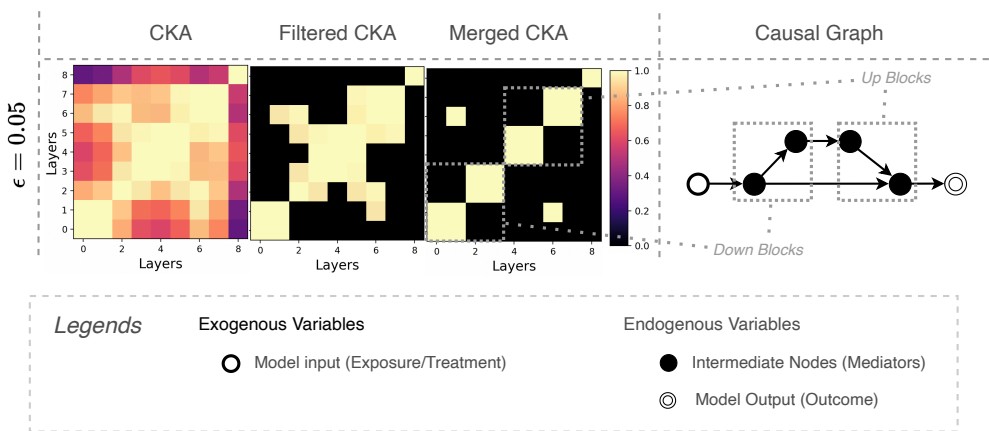

Figure 14: Causal graph inferred by TRACER for the UNet model trained on CIFAR-10. Every two blocks are merged into single causal nodes, and causal links are identified between both sequential and non-sequential nodes, reflecting the architecture's skip connections.

Specifically, to identify stochastic layers within the model, we evaluate whether any given layer produces non-consistent outputs when provided with the same input multiple times, i.e, if the outputs vary across evaluations on the same input, we classify the layer as stochastic. This allows us to detect layers where randomness influences the output, aligning with the causal interpretation of stochastic variables as external influences affecting endogenous variables within the model.

In this experiment, we infer the causal structure of a VAE trained on the MNIST dataset, focusing on its ability to reconstruct inputs and perform classification tasks. Our VAE architecture is composed of: an encoder that maps inputs to a latent space, a stochastic layer implementing the reparameterization trick, a decoder that reconstructs the data from the latent representation, and a classification layer. By explicitly modelling stochasticity as an exogenous variable, as shown in Figure 15, TRACER captures the influence of randomness on the network's behaviour, providing a more accurate representation of the causal relationships within the model.

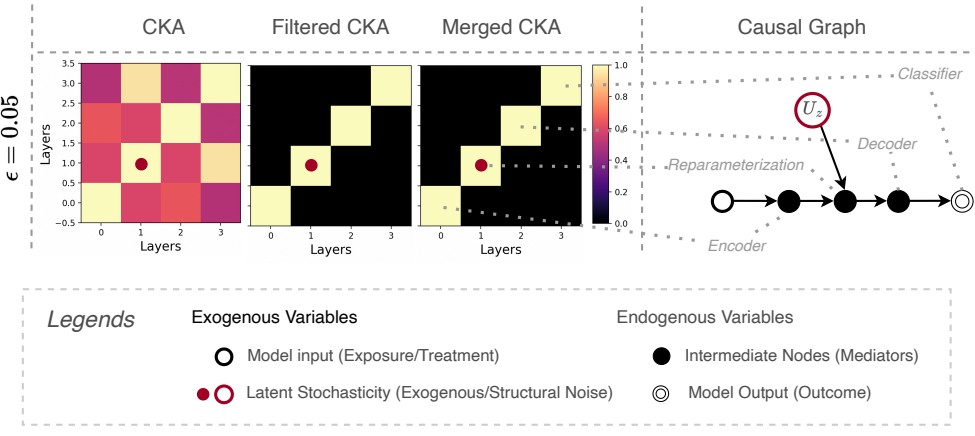

Figure 15: Causal graph inferred by TRACER for the VAE model. The stochastic layer (reparameterization) is modeled as an exogenous variable influencing the decoder and classification layers.

Our causal analysis reveals how randomness propagates through the network, affecting the decoder's output and, consequently, the final prediction. This approach allows us to interpret stochastic models by understanding how the stochastic components contribute to the overall decision-making process, highlighting the pathways through which they affect the network and helping identify areas where uncertainty may impact performance.

