# OpenReview forum: "Neural Networks Decoded: Targeted and Robust Analysis of Neural Network Decisions via Causal Explanations and Reasoning"
_ICLR.cc/2025/Conference — Submitted to ICLR 2025_

### Official Review · Reviewer_jNKN · 2024-10-16

**Soundness:** 2
**Presentation:** 1
**Contribution:** 2
**Rating:** 6
**Confidence:** 4

**Summary:**

This paper present method grounded in causal theory to estimate the causal link between the various layers of neural networks. This technique is used for both the explanation of the model and its optimization.

**Strengths:**

The use of the causal paradigm for building an explanability tool ensures it has interesting groundings.
The related work section is thorough.
A various number of considerably unique experiments are considered.

**Weaknesses:**

1. The first major weakness concerns the writing of the paper.

1.1 I see the « proofs » in section A of the appendices as explanations as to why the propositions and theorems presented in the main paper make sense and are coherent, and not formal nor rigorous proofs (especially A.1).

1.2 In section 4.1, no explanation is given about what P is in practice; the proportion p that is used. The layer grouping is explained in Section 3, but no details regarding the inputs themselves are provided. How are the most important features found? Many parts of the approach and the experiments lack clear explanations of the various steps, leading to many unanswered questions (see the Questions section) and questions found in the present section. Figure 1 does not give much information about the presented approach. How are the counterfactuals generated in practice, and how are they used? How is the feature contribution made?

1.3 There are many inconsistencies or errors in the notation and the writing itself, see ** Typos and such**.

2. « Causality » is central in the article, yet I see no causality in the proposed approach.

2.1 For example, let’s consider the way the various layers in the network are compared via Centered Kernel Alignments. This « distance measure », or similarity measure, does not tells us anything about causality, simply about resemblance.

2.2 The relationship is considered or examined between the various components of a single level (e.g. feature level), which truly undermine the scope of the analysis.

3.1 The approach is based on the fact that the considered networks are straightforward in their nature; the output of a layer is considered to be the input of the next layer, but what if more sophisticated architecture is considered, such as U-nets? How would the approach work in the case of recurrent networks? Or networks with parallel components, such as attention heads?

3.2  I am conceptually unsure of the approach, in that it only considers huge perturbations made by a single layer. What about if the whole network is a composition of many small perturbations, such as in Variational autoencoders?

3.3 It is said that if a group of layers is similar, they can be regrouped in a single transformation $g_{i,j} = fj ◦ fj−1 ◦ . . . ◦ fi = fi$, but how seeing this part of the network as a single transformation helps to reduce the number of parameters? Simply replacing $ fj ◦ fj−1 ◦ . . . ◦ fi$ by $fi$ simply cannot be done when the dimensions of the various layers aren’t the same.

4. The last concern is about the contributions of the paper. Line 532 : « Through our foundational principles and findings, we have ascertained that by producing intuitive, human-interpretable explanations, TRACER offers outstanding transparency to neural networks [...] » The paper proposes a way to compute how big of a transformation is applied at each step of the network, and to generate counterfactual, so I would not say so.

4.1 Concerning the originality of the approach: The use of counterfactual is not a contribution, for the same procedure could be used with a different model



**Typos and such**

-Line 89 : rior → Prior

-Line 96 : explainabibilty → explainability

-Line 161 : What does do() mean?
-Line 172 : Based on the notation only, it seems like it is the masked input that is in $\{0,1\}^d$, where, I guess, it is implied that it is the mask that is in $\{0,1\}^d$.

-Line 175 : Please define the « |= » operator.

-Definition 2 : In both C2 and C3, the notation is wrong, for you consider applying masks of dimension different than d to set of examples of dimension d. For example, in C3, instead of « for all M’ $\subset$ M », you probably meant « For all M’ such that |M’|₁ < |M|₁ : … ».

-Line 231 : Please provide citations for «  … prevalent approach for quantifying the
similarity between high-dimensional embeddings. »

-Line 231 : Notations are introduced for talking about the inner components of neural networks at too many various places in the article; there should be a section where this notation is properly introduced. Please state explicitly the dimensions of $f_i$ and $f_j$. It is not obvious only looking at the notation whether $K_i$ is a matrix of size $nxn$ or else.

-Line 252 : The composition increases the layer number at certain places (line 252, 259) but decreases at other places (line 259, 266). Please be coherent with the mathematical notation.

-Line 197 : The usage of « nodes » in « causal nodes » refers to the concept of layer in this work, but « node » typically refer to a single neuron of a layer. It wold be less ubiquitous to use another term for talking about « causal nodes ».

-Line 359 : when citing MNIST, please consider the following indication: http://citebay.com/how-to-cite/mnist/.

-Line 378 : The hyperparameters that were used should be found in the appendices.

-Line 408 : This letter P (with this exact calligraphy) is already used for Probability.

-Table 1 : C1, C2 and C3 already refers to concept in Definition 2. Please use distinct notation.

-Line 512 : « In this study, we focused our evaluations of TRACER on white-box neural network. » I consider neither AlexNet nor ResNet-50 to be white-box models.

**Questions:**

1. Section 3.2.1 : Why consider a single unique value to replace a subset of $x$, and why would it be different than say 0?

2. Line 241 : « Flexibility: It accommodates various kernel functions, such as linear or Gaussian, enabling flexibility based on specific requirements of the analysis ». Please explain in which situation it would be preferable to have a Gaussian kernel instead of a linear one. Why this current choice of kernel?

3. Line 287 : Why consider the many neurons of a layer as realizations of a random variable? To which probability are associated?

4. Line 292 : How does $P (g′i(x) | do(X = x′))$ and $P (g′i(x) | do(X = x))$ are computed in practice?

5. Line 408 : How does P is found in practice and what does P actually outputs? A mask?

6. Table 1 : How is the simpler model created?

---

> ### Author Response · Authors · 2024-11-22
>
> We thank the reviewer for their thorough and insightful feedback. Below, we respond to each comment and highlight specific changes made to the content of the paper.
>
>
> **Reviewer comment:**
>
> > The first major weakness concerns the writing of the paper.
>
> **Response:**
>
> We reviewed and revised the paper to eliminate any vague language, ensure that all statements are clear, and replaced subjective terms with precise language. We have also clarified the flow of the paper and the terminology used, provided better motivation and justification for the approach, and added more detailed descriptions of the experimental setups.
>
>
> **Reviewer comment:**
>
> > I see the « proofs » in section A of the appendices as explanations as to why the propositions and theorems presented in the main paper make sense and are coherent, and not formal nor rigorous proofs (especially A.1).
>
> **Response:**
>
> We acknowledge that the proofs in Appendix A lacked formal rigor. We have revised all the proofs in the Appendix to ensure they are formal, rigorous, and detailed.
>
>
> **Reviewer comment:**
>
> > In section 4.1, no explanation is given about what P is in practice; the proportion p that is used. The layer grouping is explained in Section 3, but no details regarding the inputs themselves are provided. How are the most important features found? Many parts of the approach and the experiments lack clear explanations of the various steps, leading to many unanswered questions (see the Questions section) and questions found in the present section. Figure 1 does not give much information about the presented approach. How are the counterfactuals generated in practice, and how are they used? How is the feature contribution made?
>
> **Response:**
>
> We have added in Section 4.1 of the paper (Causal Discovery and Feature Attributions) an explicit formulation of the perturbation function P (renamed as $\mathcal{P}$) and the value used for perturbation ratio p (as implemented in practice).
>
> Section 3 (Theoretical Foundations and Methodology) provides details of the framework, and the datasets used as inputs are included in Section 4 (Experiments and Results).
>
> The most important features are found based on the absolute values of their contributions (Average Causal Effects), described in Section 3.4 (Estimation of Causal Effects). Features with the highest positive ACE values constitute most important features that positively contribute to the decision, and features with the lowest negative ACE values constitute most important features that negatively contribute to the decision.
>
> The revised paper now provides clearer explanations of the experiments, and we updated Figure 1 to additionally illustrate: the input data, interventions, counterfactuals, feature attributions (explanations), and causal graphs.
>
> In practice, counterfactuals are generated using a Generative Adversarial Network (depicted in Figure 6). Due to space constraints, the experiments on counterfactual generation are now provided in the appendix. Appendix C conceptually describes the counterfactual generator, provides its implementation details, and illustrates its generation process. Additionally, Figure 7 highlights the use of counterfactuals in contrastive analysis, and Appendix F lists the hyperparameters used for this experiment.
>
>
> **Reviewer comment:**
>
> > There are many inconsistencies or errors in the notation and the writing itself, see ** Typos and such **.
>
> **Response:**
>
> We thoroughly reviewed the paper to correct inconsistencies and writing issues as indicated in ** Typos and such **, with the following exception:
>
> * The MNIST citation already follows the bibtex format provided on http://citebay.com/how-to-cite/mnist/

---

> > ### Author Response · Authors · 2024-11-22
> >
> > **Reviewer comment:**
> >
> > > « Causality » is central in the article, yet I see no causality in the proposed approach.
> > >
> > > For example, let’s consider the way the various layers in the network are compared via Centered Kernel Alignments. This « distance measure », or similarity measure, does not tells us anything about causality, simply about resemblance.
> > >
> > > The relationship is considered or examined between the various components of a single level (e.g. feature level), which truly undermine the scope of the analysis.
> >
> > **Response:**
> >
> > Our approach is fundamentally grounded in causal inference theory, specifically within the framework of structural causal models and Pearl's causal hierarchy. By intervening on input features, we effectively perform controlled "do-operations” that allow us to infer causal relationships within the neural network by examining the direct and indirect effects of specific input features on the network's behaviour.
> >
> > While Centered Kernel Alignment (CKA) is indeed a similarity measure in its own, we use it in our framework to quantify changes in layer representations resulting from causal interventions. By measuring how these interventions lead to changes in the similarity of layer activations, we can infer causal dependencies between layers. Specifically, significant changes in representation similarity, as measured by CKA, based on interventions, indicate a causal influence of the intervened input features on the activations of specific layers. This goes beyond mere resemblance, as it ties the observed changes directly to the controlled interventions, aligning with principles of causal inference.
> >
> > Our approach examines the relationships at the feature level to provide feature attributions (explaining how each feature contributes to the decision). However, the influence of these features on the internal process (layers) or the network also reveals the high-level interactions that drive the overall causal structure of the network. These two levels of analysis allow our framework to identify the feature-level dynamics, and also describe the causal mechanisms of the network at a higher, more abstract (more intuitive) level.
> >
> >
> > **Reviewer comment:**
> >
> > > The approach is based on the fact that the considered networks are straightforward in their nature; the output of a layer is considered to be the input of the next layer, but what if more sophisticated architecture is considered, such as U-nets? How would the approach work in the case of recurrent networks? Or networks with parallel components, such as attention heads?
> >
> > **Response:**
> >
> > Without making any specific assumptions about the architecture of the networks, we have successfully applied our approach to simple MLP and CNN-based models, and also models with residual connections (ResNet-50). The approach will work similarly on U-nets and recurrent networks, as we only focus on activation outputs for the causal discovery. We will aim to include such examples in the final version of the paper. Models with parallel components, such as attention mechanisms, might however require minor adjustments in our approach. We will leave the case studies for such architectures to future work.
> >
> > **Reviewer comment:**
> >
> > > I am conceptually unsure of the approach, in that it only considers huge perturbations made by a single layer. What about if the whole network is a composition of many small perturbations, such as in Variational autoencoders?
> >
> > **Response:**
> >
> > Our approach is designed to capture both large and small perturbations, and importantly, to account for the cumulative effects of small transformations across multiple layers. When we intervene on input features and observe how these interventions affect the network's activations and outputs, we are effectively analyzing how perturbations, regardless of their magnitude, propagate through the network. This allows us to capture the incremental contributions of small perturbations at each layer and understand their collective impact on the final output.
> >
> > Specifically addressing models with embedded randomness like VAEs, our approach is able to handle the inherent stochasticity by running the layers multiple times for the same input and monitoring any variations in their outputs. These detected variations are treated as exogenous variables within our causal framework, allowing us to seamlessly integrate randomness into the discovered causal mechanisms. We will add an experiment with VAEs in the final version of the paper.

---

> > > ### Author Response · Authors · 2024-11-22
> > >
> > > **Reviewer comment:**
> > >
> > > > It is said that if a group of layers is similar, they can be regrouped in a single transformation
> > > $g_{i,j} = f_j \circ f_{j-1} \circ \cdots \circ f_i = f_i$, but how seeing this part of the network
> > > > as a single transformation helps to reduce the number of parameters? Simply replacing
> > > > $f_j \circ f_{j-1} \circ \cdots \circ f_i$ by $f_i$ simply cannot be done when the dimensions of
> > > > the various layers aren’t the same.
> > >
> > > **Response:**
> > >
> > > The grouping of functionally similar layers into a single causal node simplifies the causal graph to help with interpretability by focusing on the key transformations that significantly influence the network's output.
> > > When compressing the network’s architecture based on such causal groupings, we remove redundant layers in the neural network based on their CKA similarities (CKA ≥ 0.95). Layers within the same causal node are replaced with a single representative layer where input and output sizes are adjusted to align with the rest of the network, thereby reducing the overall number of parameters.
> > > Appendix E (Global Explainability) now includes a comprehensive table comparing the baseline and compressed models, showing the architectural changes performed for each compressed version of the network.
> > >
> > > **Reviewer comment:**
> > >
> > > > The last concern is about the contributions of the paper. Line 532 : « Through our foundational
> > > > principles and findings, we have ascertained that by producing intuitive, human-interpretable explanations,
> > > > TRACER offers outstanding transparency to neural networks [...] »
> > > > The paper proposes a way to compute how big of a transformation is applied at each step of the network,
> > > > and to generate counterfactual, so I would not say so.
> > >
> > > **Response:**
> > >
> > > The contributions of our proposed framework extend significantly beyond computing transformation magnitudes at different layers of a network and generating counterfactuals.
> > > On one hand, we provide feature attributions that clearly explain how each input feature contributes to the network’s output. And on the other hand, we provide high-level causal maps that abstract the complex internal computations of neural networks into functional components and their relationships to each other, which are much more intuitive and interpretable compared to the network’s architecture.
> > > Through such causal maps, we have shown that we can derive interpretations that allow refinements such as compressing models without significant loss in accuracy.
> > >
> > > **Reviewer comment:**
> > >
> > > > Concerning the originality of the approach: The use of counterfactual is not a contribution, for the same procedure could be used with a different model
> > >
> > > **Response:**
> > >
> > > While counterfactuals have indeed been extensively explored in existing literature, our approach introduces several novel elements that distinguish it from existing methods. Specifically, we integrate a k-Nearest Neighbours (k-NN) classifier to identify real target class examples, guiding the generator to produce more realistic and class-consistent counterfactuals. Additionally, our training objective uniquely introduces constraints that ensure generated counterfactuals are plausible and realistic, while being classified as the target class.
> > >
> > > Within our causal analysis framework, the generated counterfactuals complements interventions by providing explanations that illustrate which specific changes should be made to inputs to correct misclassifications.
> > > An analysis of such counterfactuals alongside interventions, allow our framework to identify parts of the network that show significant behavioural changes due to minor differences in inputs. Further study will be required to assess the potential use of such insights in developing more robust networks.

---

> > > > ### Author Response · Authors · 2024-11-22
> > > >
> > > > **Reviewer comment:**
> > > >
> > > > > Section 3.2.1 : Why consider a single unique value to replace a subset of x, and why would it be different than say 0?
> > > >
> > > > **Response:**
> > > >
> > > > The selected value should ideally be neutral and causally independent of the features being analyzed, allowing the effect of the intervention to isolate the causal influence of those features. In game theory, particularly Shapley values, the choice of baseline value is critical because it defines the "null state" against which contributions of individual features are measured. A well-chosen baseline should ensure that the effects attributed to features are meaningful and independent of biases introduced by the baseline itself [1].
> > > >
> > > > In our experiments, we use 0 as it is a common choice [2] (e.g., 0 represents black in the images datasets and indicates absence of information in the intrusion detection dataset). By choosing a single value that is causally independent and neutral or representative of a baseline, the analysis ensures that observed effects are due to the intervention itself and not artifacts of the chosen replacement value.
> > > >
> > > > While 0 is a common choice, it is not always neutral in a causal sense. In such cases, the mean or median of a feature across the dataset can serve as a better approximation of a "neutral" or "typical" state. However, we note that using the mean or median can, in some cases, compromise the exogeneity of intervened samples, as these values are derived from the dataset itself and may carry dependencies on the feature distribution. For example, the mean might encode correlations with other variables, especially in skewed or imbalanced datasets, which could bias the causal effects. These challenges align with observations in causal reasoning [3, 4], where interventions must isolate causal influence without introducing unintended dependencies. Alternatives such as using domain-specific constants or noise-based interventions can mitigate such biases in cases where exogeneity is critical.
> > > >
> > > > [1] Sundararajan, M., & Najmi, A. (2020, November). The many Shapley values for model explanation. In International conference on machine learning (pp. 9269-9278). PMLR.
> > > >
> > > > [2] Haug, J., Zürn, S., El-Jiz, P., & Kasneci, G. (2021). On baselines for local feature attributions. arXiv preprint arXiv:2101.00905.
> > > >
> > > > [3] Pearl, J. (2009). Causality. Cambridge university press.
> > > >
> > > > [4] Janzing, D., Minorics, L., & Blöbaum, P. (2020, June). Feature relevance quantification in explainable AI: A causal problem. In International Conference on artificial intelligence and statistics (pp. 2907-2916). PMLR.

---

> > > > > ### Author Response · Authors · 2024-11-22
> > > > >
> > > > > **Reviewer comment:**
> > > > >
> > > > > > Line 241 : « Flexibility: It accommodates various kernel functions, such as linear or Gaussian, enabling flexibility based on specific requirements of the analysis ». Please explain in which situation it would be preferable to have a Gaussian kernel instead of a linear one. Why this current choice of kernel?
> > > > >
> > > > > **Response:**
> > > > >
> > > > > The Gaussian kernel is typically preferable for CKA similarity when the relationships between layer representations are highly non-linear, as it maps the data into a higher-dimensional space to capture complex relationships. However, Gaussian kernels require a hyperparameter, the bandwidth, which must be carefully tuned to appropriately capture the scale of relationships, adding computational overhead and complexity.
> > > > >
> > > > > Conversely, the linear kernel is computationally efficient and directly measures similarity in the original feature space, making it effective for understanding representational similarities when linear relationships dominate. Linear CKA works well for capturing representational similarities because it compares covariance structures between layer activations, which often align with how information is processed in neural networks, especially in settings where transformations preserve linear correlations.
> > > > >
> > > > > In our study, we chose the linear kernel primarily for its computational efficiency and its established success in prior work, such as Kornblith et al. [1], which demonstrated that linear CKA robustly identifies representational similarities across various neural network architectures and gives similar results as Gaussian kernels across most experiments. This choice allows us to scale the analysis while maintaining reliable results.
> > > > >
> > > > > [1] Kornblith, S., Norouzi, M., Lee, H., & Hinton, G. (2019). Similarity of neural network representations revisited. International Conference on Machine Learning (ICML).
> > > > >
> > > > >
> > > > > **Reviewer comment:**
> > > > >
> > > > > > Line 287 : Why consider the many neurons of a layer as realizations of a random variable? To which probability are associated?
> > > > >
> > > > > **Response:**
> > > > >
> > > > > The neurons of a layer are viewed as a random variable, with their activations over varying inputs (inputs and their intervened counterparts) treated as samples from a single probability distribution. This allows us to estimate the distribution of activations and apply statistical measures to assess changes due to interventions. By modelling layer activations in this way, we can study high-level behaviours and causal relationships by analyzing how interventions propagate through the network. This perspective aligns with methods commonly used in neural network analysis, as seen in prior work [1], which similarly leveraged statistical abstractions to study representational changes and their implications for neural computations.
> > > > >
> > > > > [1] Kornblith, S., Norouzi, M., Lee, H., & Hinton, G. (2019). Similarity of neural network representations revisited. International Conference on Machine Learning (ICML).
> > > > >
> > > > >
> > > > > **Reviewer comment:**
> > > > >
> > > > > > Line 292 : How does P( g’_i(x) | do(X = x’) ) and P( g’_i(x) | do(X = x) ) are computed in practice?
> > > > >
> > > > > **Response:**
> > > > >
> > > > > These probabilities represent the distributions of the normalized layer activations (or Layer Groups) under specific interventions. In practice, they are estimated by performing controlled interventions on the input features. For $P(g'_i(x) \mid \text{do}(X = x’))$, we replace a subset of the input features in $x$ with their intervened values $x’$, propagate this modified input through the network, and collect the output activations $g_i(x’)$. Similarly, for $P(g'_i(x) \mid \text{do}(X = x))$, we propagate the original input $x$ through the network and collect the output activations $g_i(x)$. These activations are then normalized using the softmax function (to become $g'_i(x)$ and $g’_i(x’)$) to ensure they represent valid probability distributions. The interventions are designed to isolate the causal effect of the targeted features, as detailed in Section 3.3.1 of the revised paper, where a baseline value $b$ (e.g., 0) is used to ensure that the observed changes in the distributions reflect the causal influence of the intervention.

---

> > > > > > ### Author Response · Authors · 2024-11-22
> > > > > >
> > > > > > **Reviewer comment:**
> > > > > >
> > > > > > > Line 408 : How does P is found in practice and what does P actually outputs? A mask?
> > > > > >
> > > > > > **Response:**
> > > > > >
> > > > > > In practice, $\mathcal{P}$ is computed based on the binary explanation mask $M(x)$, which identifies significant features in the input $x$. Given a perturbation proportion $p$, $\mathcal{P}$ outputs a modified binary mask $P(M(x), p)$, which selects $k = \lfloor p \cdot |M(x)| \rfloor$ significant features to perturb, sampled uniformly from the regions identified by $M(x)$. This mask is applied to the selected features in $x$, generating the perturbed input $x' = x \odot P(M(x), p)$. The output of $\mathcal{P}$ is thus the perturbed mask that isolates parts of the input features identified by the explainability method as most relevant to the model's predictions. We have clarified this in Section 4.1 (Causal Discovery and Feature Attributions), aligning it with the theoretical framework and ensuring its practical implementation is fully described.
> > > > > >
> > > > > >
> > > > > > **Reviewer comment:**
> > > > > >
> > > > > > > Table 1 : How is the simpler model created?
> > > > > >
> > > > > > **Response:**
> > > > > >
> > > > > > To create the simpler models, we use Tracer to identify and remove redundant layers in the neural network. By calculating the CKA similarities between layers, we detect layers that produce highly similar activations (CKA ≥ 0.95), indicating functional redundancy. We group these functionally redundant layers into single causal nodes (for the causal graph) and prune redundant components. Specifically, for layers within the same causal node, we replace them with a representative layer where inputs and output sizes are adjusted to align with the rest of the network, thereby reducing the overall model size.
> > > > > >
> > > > > > After pruning and merging layers based on the causal graphs for global explainability (see Appendix E), each of the three resulting compressed architectures (C1, C2, C3) is trained on the entire MNIST training set, producing classification accuracies of 99.48% for C1, 99.53% for C2, and 99.64% for C3 on the test set compared to the base model’s 99.64% accuracy.
> > > > > >
> > > > > > Appendix E (Global Explainability) now includes a comprehensive table comparing the baseline and compressed models in terms of model architectures, number of parameters, classification accuracies on the test set, and corresponding compression ratios.

---

> > > ### Author Response · Authors · 2024-11-27
> > > **Re: Tracer on DNN architectures with complex structures**
> > >
> > > The updated paper now contains an experiment with U-Nets in appendix H.1, showing an interpretable causal graph with non-adjacent links where skip-connections occur.
> > >
> > > We have also added an experiment with VAEs in appendix H.2, identifying stochasticity in layers and representing randomness as endogenous variables within the causal model.

---

> ### Comment · Reviewer_jNKN · 2024-12-02
>
> I thank the authors for taking the time to consider all of my critiques and questions one by one.
>
> > We reviewed and revised the paper to eliminate any vague language [...]
>
> Thank you for making these changes.
>
> > We have revised all the proofs in the Appendix to ensure they are formal, rigorous, and detailed.
>
> I see that they're now indeed more formal and rigorous.
>
> > We have added in Section 4.1 [...]
>
> The new details added help understand the proposed approach.
>
> > Without making any specific assumptions about the architecture of the networks [...]
>
> I understand that, even though skip connections might appear between layers, if layers are grouped as in ResNet, then the proposed approach can be used considering blocks as layers. So, in this sense, the limitation would be that the architecture requires having many of these blocks. And, to my understanding, the approach wouldn't work on ensemble methods, the final predictor is an aggregation of independent predictors.
>
> > Our approach is designed to capture both large and small perturbations [...]
>
> I meant that if the perturbation is small (or similar) for each pair of layers, then the grouping will be unsuccessful or won't assess any important relation, and the created smaller networks won't have better performances than randomly merging layers.
>
> And, concerning smaller models, no conclusion can be asserted regarding the efficiency of the approach without first comparing it to various pruning and compression methods.
>
> > The grouping of functionally similar layers into a single causal node [...]
>
> I mentioned that "Simply replacing $f_j \circ ... \circ f_i$ by $f_i$ simply cannot be done when the dimensions of the various layers aren’t the same" and indeed, in Table 2 of the Appendices, the Conv2d module dimensions for C3, C2, and C1 are modified to account for the fact that the input were of size 64 x 26 x 26 instead of 64 x 13 x 13. But, where do those new parameters come from? Are there any fine-tuning to be made?
>
>
> The current version of the paper has **many** differences from the initial submission and while it is not desirable to have that many discrepancies, the current submission is cleaner and more explicit concerning its proposed approach. I will raise my score to 6.

---

> ### Author Response · Authors · 2024-12-03
>
> We thank the reviewer for their feedback and insightful observations.
>
> **Reviewer comment:**
>
> > I understand that, even though skip connections might appear between layers, if layers are grouped as in ResNet, then the proposed approach can be used considering blocks as layers. So, in this sense, the limitation would be that the architecture requires having many of these blocks. And, to my understanding, the approach wouldn't work on ensemble methods, the final predictor is an aggregation of independent predictors.
>
> **Response:**
>
> You are correct that our approach can be applied to architectures like ResNet by considering blocks as layers, i.e., single units (causal nodes) in the analysis. While this allows us to abstract the functional components, our approach is not strictly limited to networks with many such blocks; it can also adapt to architectures with fewer or differently structured blocks by appropriately defining the functional units based on representational similarity. Therefore, even with single blocks consisting of different DNN components (e.g., convolution, linear), we can still identify and group functionally similar components based on their activation patterns in response to interventions.
>
> With our current framework focusing on single networks with interconnected layers, applying it directly to ensembles where predictors operate independently would indeed pose challenges as we rely on tracing causality through connected layers. While our approach can be used to analyze each model within the ensemble separately by considering each model in the ensemble as a separate causal mechanism, further work will be required to incorporate the aggregation mechanism into the global causal graph. This could for instance be achieved by treating the aggregation function as an additional causal node and analyzing the influence of each model's output on the final decision. We will leave this exploration to future research.
>
>
> **Reviewer comment:**
>
> > I meant that if the perturbation is small (or similar) for each pair of layers, then the grouping will be unsuccessful or won't assess any important relation, and the created smaller networks won't have better performances than randomly merging layers.
> >
> > And, concerning smaller models, no conclusion can be asserted regarding the efficiency of the approach without first comparing it to various pruning and compression methods.
>
> **Response:**
>
> Our assumption is that if any two layers produce similar values, then it **must** be that they are functionally redundant, which means that a single layer can be used to replicate their combined effect (see Theorems 1 and 2 and their proofs in Appendices A.2 and A.3, respectively). By identifying layers that consistently exhibit high representational similarity, we can merge them without compromising the network's ability to learn and generalize. If all layers of a network are deemed functionally redundant, replacing them with a single component could indeed slightly reduce performance. In such cases, we have two options:
>
> * Accepting a minor reduction in performance in exchange for significant model simplification (highly reduced computational complexity);
>
> * Choosing a more appropriate similarity thresholding value to obtain functionally different layer groups.
>
> We agree that a comparison with established pruning and compression methods would provide valuable insights. That being said, our primary aim in this work is explainability via causal analysis rather than establishing new state-of-the-art performance in model compression. Our experiments show that the insights we obtain about a model's inner workings can also serve for optimizing the size of that model.
>
>
>
> **Reviewer comment:**
>
> > I mentioned that "Simply replacing $f_j \circ f_{j-1} \circ \cdots \circ f_i$ by $f_i$ simply cannot be done when the dimensions of the various layers aren’t the same" and indeed, in Table 2 of the Appendices, the Conv2d module dimensions for C3, C2, and C1 are modified to account for the fact that the input were of size 64 x 26 x 26 instead of 64 x 13 x 13. But, where do those new parameters come from? Are there any fine-tuning to be made?
>
> **Response:**
>
> When creating the compressed models (C3, C2, and C1), we first created new layers for replacing functionally redundant ones, and configured the dimensions of each new layer to ensure compatibility with preceding and succeeding layers in the compressed model. The parameters of the compressed models, including the replacement layers, were then randomly initialized before training the network from scratch on the same training sets as the original models. In our experiments, the dimensions of each replacement layer were chosen to be closest to the dimensions of the layer with minimum number of parameters inside its layer group.

---

### Official Review · Reviewer_mpnd · 2024-11-04

**Soundness:** 2
**Presentation:** 3
**Contribution:** 2
**Rating:** 6
**Confidence:** 2

**Summary:**

The paper proposes TRACER, a method for generating "explanations" of neural network behavior with methods from causal inference. TRACER groups neural network layers into "groups" based on their similarity, and then computes a causal effect score for each input dimension in terms of how that feature affects the intermediate network layers and the final output. This is done by making a series of interventions to the input and observing how that affects downstream layer groups and the network output. The authors claim that the method generates more sensible saliency maps than existing methods, as measured by a "reliability score" that they define. They also claim that their method can be used to compress an MNIST classifier by over 99% without harming accuracy.

**Strengths:**

* The paper is well-presented with high-quality figures
* The research direction, of using causal inference for explainability, is an interesting and important one.

**Weaknesses:**

* The paper uses a lot of vague and flowery language that makes me worry that it may be substantially LLM-written.
* The authors claim that they can reduce the size of an MNIST classifier by 99.42% without substantially harming model performance. This seems too good to be true, and their method for doing this not very clearly described.

Overall, I am having a hard time assessing the method and claims of the paper, so will give a score of 5 for now with low confidence.

**Questions:**

* You say that you use a pre-trained AlexNet model for MNIST classification, however AlexNet was an ImageNet model. Did you fine-tune it with a new head to classify MNIST digits? If so, why isn't this described anywhere in the paper?

---

> ### Author Response · Authors · 2024-11-22
>
> We thank the reviewer for their valuable feedback, and address each of their comments below, outlining specific changes we made to improve the paper accordingly.
>
>
> **Reviewer comment:**
>
> > The paper uses a lot of vague and flowery language that makes me worry that it may be substantially LLM-written.
>
> **Response:**
>
> We reviewed and revised the paper to eliminate any vague language, ensure that all statements are clear, and replace subjective terms with precise language. Examples of the changes made include:
>
> * “counterfactuals offer both an intuitive understanding of model decisions and actionable insights for model enhancement”\
>   replaced with\
>   “counterfactuals provide a means to understand model decisions and identify changes to input features that can correct misclassifications”.
>
>
> * “By systematically identifying how specific input features and intermediate layer activations influence the model’s final predictions, TRACER provides a unique approach, based on the principles of PCH, for capturing an abstract overview of the distinct computational components driving DNN decisions. This structured approach allows us to produce \textbf{explanations} that clarify both the direct influence of features and how the model’s predictions would change under different input conditions, thus providing a more comprehensive understanding of the decision-making process.”\
>   replaced with\
>   “By identifying how specific input features and intermediate layer activations influence the model’s final predictions, TRACER provides a unique approach for capturing an abstract overview of the distinct computational components driving DNN decisions. This structured approach allows us to produce explanations that clarify both the direct influence of features and how the model’s predictions would change under different input conditions.”
>
>
> * By systematically collecting intermediate and final outputs of the classifier, given an input sample and its interventions, TRACER enables a focused comparison of representations across network layers and extrapolates an accurate estimation of the causal dynamics driving the network's decisions.”\
>   replaced with\
>   “Given an input sample and its interventions, TRACER collects the intermediate and final outputs of the classifier to perform a focused comparison of representations across network layers and extrapolate an accurate estimation of the causal dynamics driving the network's decisions.”
>
> **Reviewer comment:**
>
> > The authors claim that they can reduce the size of an MNIST classifier by 99.42% without substantially harming model performance. This seems too good to be true, and their method for doing this not very clearly described.
> >
> > Overall, I am having a hard time assessing the method and claims of the paper, so will give a score of 5 for now with low confidence.
>
> **Response:**
>
> We use Tracer to identify and remove redundant layers in the neural network. By calculating the CKA similarities between layers, we detect layers that produce highly similar activations (CKA ≥ 0.95), indicating functional redundancy. We group these functionally redundant layers into single causal nodes (for the causal graph) and prune redundant components. Specifically, for layers within the same causal node, we replace them with a representative layer where input and output sizes are adjusted to align with the rest of the network, thereby reducing the overall model size.
>
> After pruning and merging layers based on the causal graphs for global explainability (see Appendix E), each of the three resulting compressed architectures (C1, C2, C3) is trained on the entire MNIST training set, producing classification accuracies of 99.48% for C1, 99.53% for C2, and 99.64% for C3 on the test set compared to the base model’s 99.64% accuracy. This produces 11,696,202 total parameters for the base model and 66,218 total parameters for C1.
>
> Appendix E (Global Explainability) now also includes a comprehensive table comparing the baseline and compressed models in terms of model architectures, number of parameters, classification accuracies on the test set, and corresponding compression ratios.
>
>
> **Reviewer comment:**
>
> > You say that you use a pre-trained AlexNet model for MNIST classification, however AlexNet was an ImageNet model. Did you fine-tune it with a new head to classify MNIST digits? If so, why isn't this described anywhere in the paper?
>
> **Response:**
>
> Sorry about the confusion.
>
> We used the AlexNet architecture as the base model for our MNIST experiments. Since AlexNet is originally designed for ImageNet classification, we modified it to take as input single-channel images of size 28×28 pixels, and changed the output layer to have 10 units corresponding to the MNIST classes. The base model was then trained from scratch on the MNIST training set.
>
> This correction and clarification is now made in the introductory paragraph of Section 4 (Experiments and Results).

---

> > ### Comment · Reviewer_mpnd · 2024-11-27
> > **Thanks for addressing my concerns**
> >
> > > We reviewed and revised the paper to eliminate any vague language, ensure that all statements are clear, and replace subjective terms with precise language.
> >
> > Thank you for making these changes.
> >
> > > Appendix E (Global Explainability) now also includes a comprehensive table comparing the baseline and compressed models in terms of model architectures, number of parameters, classification accuracies on the test set, and corresponding compression ratios.
> >
> > Thank you for including this info now, that's helpful.
> >
> > > We used the AlexNet architecture as the base model for our MNIST experiments. Since AlexNet is originally designed for ImageNet classification, we modified it to take as input single-channel images of size 28×28 pixels, and changed the output layer to have 10 units corresponding to the MNIST classes. The base model was then trained from scratch on the MNIST training set.
> >
> > Okay that makes sense. And thanks for clarifying that in Section 4 now.
> >
> > This addresses the main concerns I had about the paper so I will raise my score to a 6, but still with low-confidence.

---

### Official Review · Reviewer_hQkT · 2024-11-05

**Soundness:** 3
**Presentation:** 3
**Contribution:** 3
**Rating:** 6
**Confidence:** 4

**Summary:**

In this paper, the authors propose a novel method, called TRACER, to investigate the causal dynamics inside DNNs by intervening on input features. They further use TRACER to generate counterfactuals to improve explainability.

**Strengths:**

- The paper is well organized and well written.

- The proposed method is technically sound.

- The problem is of great importance and of interest to the community

**Weaknesses:**

- As we know that DNNs are strong correlation learners, the learned links/parameters between neurons/layers might contain lots of strong spurious connections, rather than causal connections. In this case, we cannot distinguish them even though intervening on input features and monitoring the changes on intermediate outputs. Thus, I guess that it might be difficult to learn a true causal graph.

- Many details in the experimental part are missing, e.g., what interventions are performed in each specific experiment? What specific regularization metric is chosen for the counterfactual analysis? etc.

**Questions:**

- It seems not to show a counterfactual generation for a misclassified ImageNet sample? I want to see the quality of such a bit complicated counterfactual.

- It is well known GANs have the mode collapse problem. Does it occur in your experiments? If so, how to deal with it?

---

> ### Author Response · Authors · 2024-11-22
>
> We thank the reviewer for their insightful and constructive feedback. Below, we address each comment in detail and outline specific changes we made to improve the paper accordingly.
>
>
> **Reviewer comment:**
>
> > As we know that DNNs are strong correlation learners, the learned links/parameters between neurons/layers might contain lots of strong spurious connections, rather than causal connections. In this case, we cannot distinguish them even though intervening on input features and monitoring the changes on intermediate outputs.
> Thus, I guess that it might be difficult to learn a true causal graph.
>
> **Response:**
>
> Indeed, distinguishing true causal relationships from spurious correlations in DNNs is a significant challenge.
>
> However, our method is designed with this issue in mind, employing multiple interventions and leveraging Centered Kernel Alignment (CKA) analysis. By systematically intervening on input features and observing the impact on intermediate activations across various inputs, we can discern consistent causal patterns and minimize the influence of spurious correlations.
> Spurious correlations, which may arise due to coincidental associations in the data, are less likely to affect the activations across all interventions. Therefore, when we perform multiple interventions and observe that the high CKA similarities between certain layers persist, this indicates a consistent correlation.
>
> We provide a proof in the appendix, demonstrating how using multiple interventions for the CKA analysis minimizes the effects of spurious correlations.
>
>
> **Reviewer comment:**
>
> > Many details in the experimental part are missing, e.g., what interventions are performed in each specific
> experiment? What specific regularization metric is chosen for the counterfactual analysis? etc.
>
> **Response:**
>
> We agree that providing more experimental details would enhance the clarity of our paper and we have made the following changes:
>
> * Interventions performed in each experiment:
>
> Section 4.1 (Causal Discovery and Feature Attributions): […] We perform interventions by systematically masking pixels in a grid pattern, setting pixel values to zero before normalization. For MNIST, the interventions applied consist of 3x3 patches with a 1x1 sliding window. […]
>
> Section 4.2 (Generalization and Scalability): […] Intervention applied on ImageNet samples occlude parts of the inputs (setting them to zero before normalization), with 7x7 patches and a 3x3 sliding window. And for the Network Intrusion Detection dataset, we use 1$x3 patches with a 1x1 sliding window. […]
>
>
> * Regularization Metric in Counterfactual Analysis:
>
> Section 3.5 (Counterfactual Generation): […] We choose in our experiments the $\ell_1$ norm as regularization metric to encourage sparsity in the differences between $x^*$ and $x_{\text{nn}}$, promoting minimal and interpretable changes to the original input. […]
>
> * Implementation Details:
>
> Appendix F includes all hyperparamters used in our experiments, including the balancing coefficient $\lambda$ between the GAN loss and the distance regularization loss (set to 0.1), the number of epochs for training the counterfactual generator for MNIST and ImageNet (set to 15 and 100, respectively), and so on.
>
>
> **Reviewer comment:**
>
> > It seems not to show a counterfactual generation for a misclassified ImageNet sample? I want to see the quality of such a bit complicated counterfactual.
>
> **Response:**
>
> Including counterfactual examples for misclassified ImageNet samples would indeed strengthen the paper and demonstrate the applicability of our counterfactual generation method to more complex data.
>
> To this end, in the final version of the paper we will add a few such examples in the appendix, including: the original misclassified image, their generated counterfactuals, and the differences between them.
>
>
> **Reviewer comment:**
>
> > It is well known GANs have the mode collapse problem. Does it occur in your experiments? If so, how to deal with it?
>
> **Response:**
>
> This is an important concern, as mode collapse can affect the diversity and quality of generated counterfactuals. While this is a common issue in GANs, our counterfactual generation method already addresses this issue with the use of additive perturbations in the latent space before decoding (see the remark in Section 3.4 (Counterfactual Generation), in which we have updated the last sentence to clarify this point: “By introducing perturbations $\delta_i$ to the latent representation $z_x$ before decoding, training with this regularization enables the generation of multiple distinct plausible counterfactuals $x^* = D([z_x + \delta_i; o(y^*)])$, thereby reducing mode collapse”.

---

> > ### Author Response · Authors · 2024-11-27
> > **Re: Counterfactual examples for complex images**
> >
> > The revised paper now includes in Appendix C (Figure 7) instances of misclassified images, their generated counterfactuals, and the differences between the two, for samples of both the MNIST and CIFAR-10 datasets.

---

### Official Review · Reviewer_w9Xg · 2024-11-06

**Soundness:** 2
**Presentation:** 2
**Contribution:** 2
**Rating:** 6
**Confidence:** 2

**Summary:**

This paper (TRACER) aims to analyze the causal dynamics of deep neural network (DNN) decisions without altering their architecture. By intervening on input features, TRACER tries to identify feature importance and constructs a causal map.

**Strengths:**

- aims to  estimate the causal mechanisms underpinning DNN decisions

- aims to provide explanations for correct and misclassified samples

**Weaknesses:**

-  The paper is poorly structured and difficult to follow, with several instances of sloppiness. For instance, in lines 157-160, it says the first level of PCH is Association; However, it is typically Abduction, which involves inferring exogenous noises.


>Association: We extract dependency structures from the DNN activations and outputs $P(Y^{(i)}| X)$
where $X$ and $Y^{(i)}$ represent the input and the $i$-th layer’s output variables, respectively;

Is $Y^{(i)}$ exogenous noise?  **Please clarify their interpretation of the Association level and how it relates to abduction**

- A clear description of causal variables is missing.
  >line (204-213) : Assuming b to be causally independent (e.g., binary mask), all input
features, before and after interventions, can be considered exogenous variables in the causal map.

How? **Please justify.**

- Many definitions and theorems lack adequate motivation. For instance, the reasoning behind the Average Causal Effect presented in that specific form is unclear. **Please provide more context or motivation for key definitions and theorems, particularly the Average Causal Effect.**

- >*Upon obtaining the similarity measures, we establish causality by grouping layers based on their CKA values* line (246-251)...causal explanation depends on the grouping based on predetermined threshold $\epsilon$.

Different thresholds will give different causal structures or explanations for the DNN. The consistency of the explanations is not discussed.
**How does the choice of threshold affect the stability and reliability of causal explanations?**

- There is no clear explanation for selecting $\epsilon$, which is a critical hyperparameter. **How do you select or tune the
 parameter? Please  include any empirical studies or theoretical justifications for your choice**

- Figure 1 does not adequately illustrate the methodology or framework.

-  misses relevant literature, such as:
> - *Neural network attributions: A causal perspective*, Aditya Chattopadhyay, Piyushi Manupriya, Anirban Sarkar, and Vineeth N Balasubramanian.
> - *Towards learning and explaining indirect causal effects in neural networks*,  Abbaavaram Gowtham Reddy, Saketh Bachu, Harsh Nilesh Pathak, Ben Godfrey, Vineeth N. Balasubramanian, V Varshaneya, and Satya Narayanan Kar.

**A comparison with these works is necessary..**

**Questions:**

Please see the weaknesses...

---

> ### Author Response · Authors · 2024-11-22
>
> We thank the reviewer for the detailed and constructive feedback. Below, we address each of the comments and outline the specific changes we have made to the paper to address them.
>
>
> **Reviewer comment:**
>
> >The paper is poorly structured and difficult to follow, with several instances of sloppiness. For instance, in lines 157-160, it says the first level of PCH is Association; However, it is typically Abduction, which involves inferring exogenous noises.
> > > Association: We extract dependency structures from the DNN activations and outputs P(Y^{(i)} | X) where X and Y^{(i)} represent the input and the i-th layer’s output variables, respectively;
> >
> >Is Y^{(i)} exogenous noise? Please clarify their interpretation of the Association level and how it relates to abduction
>
> **Response:**
>
> We have added a paragraph at the end of the introduction section outlining the flow of the paper.
>
> The terminology used for our three levels of analysis follow closely Pearl’s framework [1], in which the first level is “Association”, not “Abduction”. Association refers to identifying statistical dependencies between variables, while Abduction refers to reasoning backward from observations to infer latent causes or exogenous variables. While abduction can be important in causal reasoning, it is not one of the levels in Pearl's Causal Hierarchy but rather a component of causal inference processes [2].
> We have added a footnote referencing the origin of the terminology used.
>
> In our framework, $Y^{(i)}$ represents layer activations, which are **endogenous variables** produced within the network and are dependent on prior computations. Therefore, these are not exogenous noise. Exogenous variables refers to factors external to the system: factors that are not modeled within the DNN activations.
>
> [1] Judea Pearl and Dana Mackenzie. “The Book of Why: The New Science of Cause and Effect” (2018). The three levels are discussed in Chapter 1, titled "The Ladder of Causation".
>
> [2] Plecko, Drago, and Elias Bareinboim. "A causal framework for decomposing spurious variations." Advances in Neural Information Processing Systems 36 (2024).
>
>
> **Reviewer comment:**
>
> >A clear description of causal variables is missing.
> > > line (204-213) : Assuming b to be causally independent (e.g., binary mask), all input features, before and after interventions, can be considered exogenous variables in the causal map.
> >
> >How? Please justify.
>
> **Response:**
>
> In our causal model, the input features are considered exogenous variables because they are not caused by any variables within the model, i.e., they are external inputs that initiate the causal processes within the neural network. When we perform interventions by modifying input features (e.g., setting them to a constant value $b$), we are manipulating these exogenous variables directly.
>
> By assuming $b$ to be causally independent (its value not depending on other variables), we ensure that the intervention does not introduce new causal dependencies within the model. Therefore, both the original and intervened input features remain exogenous, as their values are set externally and are not influenced by other variables in the model.
>
> In the paper, Definition 1 provides explanations for all types of causal variables used in our framework. Additionally, exogenous variables (lines 219-221) are defined as input features treated as independent external influences, with causal independence assumed through interventions (e.g., binary masking). This treatment isolates their effects, allowing them to function as external drivers in the causal map. Endogenous variables, as described in lines 264-266, arise from internal model dynamics, where layer activations contribute to shared causal nodes, representing variables dependent on the relationships between layers. These endogenous variables describe distinct structural equations in the causal model, capturing how internal computations propagate causal effects.
>
> We have elaborated on the role of the baseline value in the paper, justifying why input features, before and after interventions, are considered exogenous variables in our causal model.

---

> > ### Author Response · Authors · 2024-11-22
> >
> > **Reviewer comment:**
> >
> > >Many definitions and theorems lack adequate motivation. For instance, the reasoning behind the Average Causal Effect presented in that specific form is unclear. Please provide more context or motivation for key definitions and theorems, particularly the Average Causal Effect.
> >
> > Response:
> >
> > We appreciate the feedback and agree that additional context would enhance the understanding of our definitions and theorems.
> >
> > The Average Causal Effect (ACE) is a fundamental concept in causal inference, used to quantify the expected change in an outcome variable resulting from an intervention on a causal variable.
> >
> > In our framework, we adapt this metric to quantify the impact of interventions on the neural network's outputs at different layers. By calculating the ACE using the signed divergence between the layer outputs before and after the intervention, we capture both the magnitude and direction of the change induced by the intervention. This formulation allows us to understand both how (direction: positive vs negative contribution) and by how much (magnitude) the interventions influence the outputs.
> >
> > The KL divergence is a natural choice for measuring how one probability distribution diverges from another, providing insight into the informational change induced by interventions. For example, if an intervention causes a significant shift in the activation patterns at a particular layer, the KL divergence will be large, indicating a strong causal effect. Conversely, minimal changes will result in a small ACE, suggesting that the intervened features have little influence at that layer.
> >
> >
> > **Reviewer comment:**
> >
> > >> Upon obtaining the similarity measures, we establish causality by grouping layers based on their CKA values line (246-251)...causal explanation depends on the grouping based on predetermined threshold $\epsilon$.”
> > >
> > > Different thresholds will give different causal structures or explanations for the DNN. The consistency of the explanations is not discussed. How does the choice of threshold affect the stability and reliability of causal explanations?
> >
> > **Response:**
> >
> > Indeed, the threshold $\epsilon$ for grouping layers based on their Centered Kernel Alignment (CKA) values can affect the resulting causal structures. This threshold determines the level of similarity required for layers to be considered functionally similar and grouped into a single causal node.
> >
> > We have therefore expanded the discussion of this threshold to address how varying its value affects the grouping and the resulting causal graph. We explain that while a smaller $\epsilon$ (i.e., stricter similarity criteria) leads to more granular grouping, a larger $\epsilon$ results in broader grouping.
> >
> > However, this does not affect the stability or reliability of the causal explanations themselves, as the ACE is formulated to operate based on changes at the level of individual layers and final outputs of the network.
> >
> >
> > **Reviewer comment:**
> >
> > > There is no clear explanation for selecting $\epsilon$, which is a critical hyperparameter. How do you select or tune the parameter? Please include any empirical studies or theoretical justifications for your choice
> >
> > **Response:**
> >
> > In our paper, we use an $\epsilon$ of 0.05 to align with the widely accepted convention in statistical hypothesis testing, as used for the jackknife z-test in the CKA paper [1]. The p-value threshold of 0.05 has been established as a default in many scientific fields, including machine learning and deep learning research.
> >
> > By setting the similarity threshold to $1 - \epsilon = 0.95$, we ensure that only layers with representations that are almost identical (indicating they perform redundant or very similar computations) are grouped into the same causal node. From a theoretical standpoint, setting $\epsilon$ too low may fragment the network into too many causal nodes, reducing interpretability. Conversely, setting $\epsilon$ too high may over-aggregate layers, potentially merging dissimilar activations and obscuring important distinctions.
> >
> > As experiments take time, we plan to include in the appendix of the final version of the manuscript an ablation study on varying $\epsilon$ from 0 to 1 in increments of 0.05, to assess its effect on the resulting causal graphs.
> >
> > [1] Kornblith, S., Norouzi, M., Lee, H., & Hinton, G. (2019). Similarity of Neural Network Representations Revisited. International Conference on Machine Learning (ICML)
> >
> >
> > **Reviewer comment:**
> >
> > >Figure 1 does not adequately illustrate the methodology or framework.
> >
> > **Response:**
> >
> > We have updated Figure 1 to additionally illustrate: the input data, interventions, counterfactuals, feature attributions (explanations), and causal graphs.

---

> > > ### Author Response · Authors · 2024-11-22
> > >
> > > **Reviewer comment:**
> > >
> > > > misses relevant literature, such as:
> > > > * Neural network attributions: A causal perspective, Aditya Chattopadhyay, Piyushi Manupriya, Anirban Sarkar, and Vineeth N Balasubramanian.
> > > > * Towards learning and explaining indirect causal effects in neural networks, Abbaavaram Gowtham Reddy, Saketh Bachu, Harsh Nilesh Pathak, Ben Godfrey, Vineeth N. Balasubramanian, V Varshaneya, and Satya Narayanan Kar.
> > > >
> > > > A comparison with these works is necessary.
> > >
> > > **Response:**
> > >
> > > Thank you for these pointers.
> > >
> > > Our approach differs from these methods in that we focus on modelling causal relationships not just between input features and outputs but also among internal layers of the network. By grouping layers into causal nodes based on CKA similarities, we provide a hierarchical and interpretable representation of the network's decision-making process.
> > >
> > > We have included in the related work section of the paper how our approach differs from and complements these works.

---

> > > ### Author Response · Authors · 2024-11-27
> > > **Re: Empirical study for selecting $\epsilon$**
> > >
> > > The revised paper now includes in Appendix G an ablation study on the impact of $\epsilon$ (taking values {0, 0.05, 0.1, 0.2, 1}) on the causal discovery process.

---

> ### Comment · Reviewer_w9Xg · 2024-12-03
>
> Thanks for the rebuttal. I have revised my score.

---

### Meta-Review · Area_Chair_MCki · 2024-12-23

**Metareview:**

This paper analyzes neural networks through causal dynamics for improved explainability by making interventions, defining similarity metrics between layers, and also counterfactuals through a GAN. Two of the reviews had low confidence and the final review was short, but positive.  The confident reviewer raised some issues that were partially addressed by the authors in the rebuttal about counterfactual examples. These were added to an appendix.

Given this context, I took a close read of the paper myself and found several concerns:

1) The paper is hard to follow mostly because many definitions and theorems lack adequate motivation (agreeing with Reviewer w9Xg)
2) For input level attribution, there's a pretty big literature on explanation by removing that was not acknowledged and also a literature on how to evaluate them
* Explanation by Removing
   - Explaining by Removing: A Unified Framework for Model Explanation by Covert et al (https://arxiv.org/abs/2011.14878)
   - Have We Learned to Explain?: How Interpretability Methods Can Learn to Encode Predictions in their Interpretations (https://arxiv.org/abs/2103.01890)
 * Evaluations for input level attribution
   - A Benchmark for Interpretability Methods in Deep Neural Networks (https://arxiv.org/abs/1806.10758)
   - Explanations that reveal all through the definition of encoding (https://arxiv.org/abs/2411.02664)

3) Counterfactual were discussed using a GAN, but they weren't evaluated. This was fixed, but placed in an appendix. It also wasn't obvious why something like that was needed because the structural causal equation (the model) is known, moreover, the outputs do not have any extra randomness that could be conditioned on for some type of counterfactual.

4) The writing of the paper could be improved. The reviewers liked the quality of the figures, but thought the text was often vague and hard to follow (from my own read, I agree with this assessment).

5) Too many pointers to the appendix (which looks like it has some interesting results)

The positive part of the paper was on the coalescing of layers using the CKA. Overall, I think there's merit in the work, but needs a good amount of refinement before it's ready for publication.

**Additional Comments On Reviewer Discussion:**

All the reviewers agreed and were positive about this paper. However, two of the reviewers had low confidence.  The reviewers liked the quality of the figures, but thought the text was often vague and hard to follow from two of the reviewers. The confident reviewer raised some issues that were partially addressed by the authors in the rebuttal about counterfactual examples. The clarity issues around definitions raised by reviewer w9Xg were replied to, but from my own read of the latest version more needed to be done to improve definitions. Similar issues were raised by reviewer mpnd and were replied to by the authors, but I also felt more needed to be done to make the paper transparent and crisp.

---

### Decision · Program_Chairs · 2025-01-22

Reject